# An Efficient Framework for Length Extension via Dynamically Growing Positional Embedding and Routing Attention

## Abstract

Modeling long sequences is critical for numerous large-scale models. However, extending existing architectures to handle significantly longer sequences poses substantial technical and computational challenges. One inevitable issue is the overfitting of large models to positional encodings during pretraining, which limits their ability to generalize to unseen positional encoding scales. Additionally, extending sequence lengths requires extensive computational resources and time. Existing positional encoding methods often rely on carefully designed scaling factors but typically yield suboptimal results. To tackle these challenges, we propose **Cyclic, Randomly Truncated, and Dynamically Growing NTK Positional Embedding (CRG NTK)**, a data-augmentation-based technique that fully explores the RoPE encoding space, enabling models to adapt to various positional scales and achieve state-of-the-art extrapolation for the extension of lengths dominated by position encoding. Furthermore, we introduce **an efficient attention mechanism with a correlation-based routing strategy to enhance the fitting of the augmented positional encoding**, yielding superior performance and more efficient fine-tuning. With our approach, LLaMA-7B and Mistral-7B fine-tuned at 16K context length achieve extrapolation factors of at least $128\times$ on simple tasks and maintain stable perplexity over $32\times$ sequence length extensions and saves at least 16 times the GPU training resources compared to the existing optimal method. Experiments also show that correlation routing can achieve good performance by further filtering out large amounts of noise in long sequences.

## 1 Introduction

In various natural language processing (NLP) tasks, such as document-level sentiment analysis Behdenna et al. (2018), long document summarization Koh et al. (2022), and code generation Rozière et al. (2024), the ability to effectively model long-sequence dependencies is crucial. This capability allows for capturing complex relationships over sequences spanning hundreds or thousands of tokens, which is essential for tasks where contextual information is dispersed. Consequently, extending the context window enables large language models (LLMs) to perform tasks that shorter context windows cannot handle and potentially enhances performance across a variety of NLP tasks.

However, extending the context window of LLMs Touvron et al. (2023a); Jin et al. (2023) is a critical research direction, yet it faces numerous challenges. Foremost among these is the influence of positional encoding mechanisms on the context window extension. During pre-training, models typically operate within fixed-length windows using predefined positional encodings. This often leads to overfitting to the positional distribution within the training range, thereby limiting the model's ability to handle unseen positional information beyond this range. Furthermore, the resource requirements for extending the context window are significant, involving not only substantial storage space but also excessive computation time due to the need for fine-tuning on extremely long sequences.

To address the limitations of positional encodings, recent work Press et al. (2022); Chi et al. (2022); Li et al. (2024); Zheng et al. (2024a) has extensively analyzed relative mechanisms, particularly Rotary Positional Embedding (RoPE) Su et al. (2023). A common approach to extending RoPE involves scaling positional

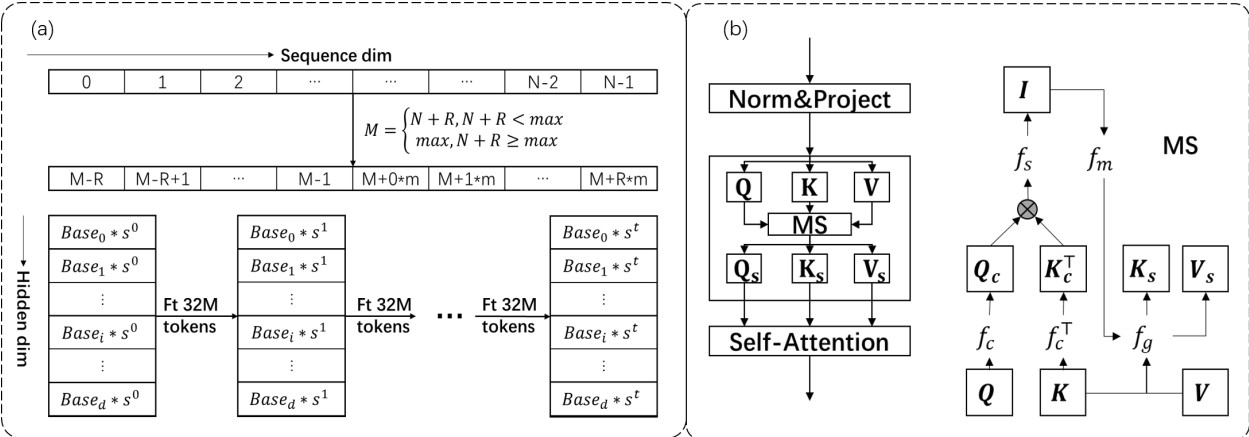

Figure 1: (a) During fine-tuning, the CRG NTK (Cyclic, Randomly Truncated, and Dynamically Growing NTK Positional Embedding) mechanism is employed for context length extension. $N$: Original sequence length. $R$ and $m = 1$: Randomly sampled starting index for cyclic truncation. $s^k$: scale for Dynamically Growing. "Ft 32M tokens": fine-tuning with 32 million tokens. (b) Efficient fine-tuning architecture using Merge and Select Attention (MS Attention) integrated with LoRA. $I$: Indexes for selection. $f_s, f_m$: Select and merging functions. $f_c, f_c^T, f_g$: Compress and gather functions in MS Attention.

indices, effectively mapping out-of-distribution positions back into the pre-trained range. For example, Positional Interpolation (PI) Chen et al. (2023) rescales indices to fit within the original context, while NTK-aware Ntk (2023) methods scale the base frequency to modestly enhance extrapolation. YaRN Peng et al. (2023) improves on this by separating frequencies into high, mid, and low bands, applying different scaling strategies: leaving high frequencies unchanged, scaling low frequencies with PI, and adjusting mid frequencies using NTK-aware methods. LongRoPE Ding et al. (2024) further optimizes scaling ratios and anchor positions through a search algorithm. Despite these efforts, most methods rely on carefully crafted scaling heuristics and still fall short in capacity for extrapolation and performance. Moreover, context extension also faces hardware bottlenecks: large model sizes demand significant compute and memory. To mitigate this, LongLoRA Chen et al. (2024) integrates efficient Attention, DeepSpeed Rasley et al. (2020), and LoRA Hu et al. (2021) to reduce fine-tuning time and storage requirements.

To overcome these limitations, we propose a data-augmentation-based positional encoding method designed to improve the generalization ability of positional encodings. Our approach employs dynamically growing scaling ratios, allowing positional encodings to scan a broader solution space to achieve stronger extrapolation capability. Based on the above method, we provide a degree of theoretical interpretation and empirical support for the extrapolation capacity of models: **the extrapolation limit is approximately bounded by the model depth.** This extrapolation-based context extension strategy not only significantly reduces the resource requirements for expanding the model's context length, but also substantially advances the study of foundation models, such as understanding the transmission of information across layers.

We further optimize the LoRA configuration and design an efficient attention mechanism aligned with positional encoding strategies. Our relevance-based routing selectively weights value tokens based on relevance-based routing mechanism, reducing computation while enhancing the model's dependence on accurate positional cues. This allows effective regression even with partial positional information. We argue that simply increasing sequence length does not unconditionally improve model performance. As the sequence length grows, the beneficial information gain diminishes, while the negative impact of noise becomes increasingly pronounced. Although Softmax provides limited noise suppression, it is inadequate for long sequences. In contrast, our routing mechanism explicitly filters irrelevant tokens, enabling efficient attention under controlled noise and improving long-sequence performance.

In summary, our method tackles the core challenges of context window extension by introducing advances in both positional encoding and efficient Attention. Through theoretical analysis and empirical validation, we demonstrate superior extrapolation capabilities and performance. Leveraging CRG NTK positional encoding,

we achieve extrapolation up to $32\times$ the fine-tuning length on a 32-layer model—approaching the hypothesized limit. Then, combining with MS Attention our method surpasses LongRoPE with at least $16\times$ less GPU fine-tuning time and comparable performance. Furthermore, our approach achieves over $128\times$ extrapolation on simple retrieval tasks. These results offer new insights and practical strategies for efficiently scaling context windows in LLMs.

## 2  Preliminary

### 2.1  Transformer

The Llama-2 Touvron et al. (2023b) and Mistral Jiang et al. (2023) models are based on the Transformer architecture, which consists of the core modules: self-attention and feed-forward network (FFN). The computation process of self-attention Vaswani et al. (2017) is as follows:

$$O = \mathrm{softmax}(QK^\top)V, \tag{1}$$

where $Q$, $K$, and $V$ are obtained from the input $X \in \mathbb{R}^{n\times d}$ using embedding weights $W_q$, $W_k$, and $W_v$, respectively. The final output $O$ is then passed through $W_o$ to obtain the final output of the attention layer. Subsequently, the entire Transformer layer is completed through the FFN.

#### 2.1.1  Fixed-Pattern and Dynamically Routed Sparse Attention

This category of methods restricts queries ($Q$) to fixed or sequence-independent key ($K$) positions, which significantly reduces the computational cost. The general formulation is:

$$\tilde{K} = \mathrm{Loc}(K, i_k(Q,K)), \quad \tilde{V} = \mathrm{Loc}(V, i_k(Q,K)), \quad \mathrm{output} = \mathrm{Softmax}(Q\tilde{K}^\top)\tilde{V}, \tag{2}$$

where $\mathrm{Loc}(\cdot)$ represents a selection function, and $i_k$ denotes the selected indices. In fixed-pattern approaches, $i_k$ is independent of $Q$ and $K$. Let $w_{\mathrm{win}}$ be the sliding window size and $l$ be the shift size:

- **Sliding Window:** $i_k \in (i - \frac{w_{\mathrm{win}}}{2}, i + \frac{w_{\mathrm{win}}}{2})$.

- **Global Tokens:** $i_k \in (0, w_{\mathrm{win}})$, commonly used in attention sink mechanisms.

- **Swin Transformer** Liu et al. (2021): alternates between windowed attention $i_k \in (i - \frac{w_{\mathrm{win}}}{2}, i + \frac{w_{\mathrm{win}}}{2})$ and shifted window attention $i_k \in ((i - \frac{w_{\mathrm{win}}}{2} + l) \bmod n, (i + \frac{w_{\mathrm{win}}}{2} + l) \bmod n)$.

- **BigBird** Zaheer et al. (2021): incorporates sliding window, global tokens, and randomly selected keys and values: $i_k \in (r_1, r_2, \ldots, r_m)$.

Unlike fixed-pattern approaches, dynamically routed methods determine key ($K$) positions based on the importance of different regions, thereby achieving improved performance. In other words, the selected indices $i_k(Q,K)$ depend on $Q$ and $K$.

For instance, BiFormer Zhu et al. (2023) selects keys dynamically using:

$$i_k(Q,K) = \arg\max_{\mathrm{top}\text{-}K}(\bar{Q}\bar{K}^\top), \tag{3}$$

where $\bar{Q} \in \mathbb{R}^{n/w_{\mathrm{win}}\times d}$ and $\bar{K} \in \mathbb{R}^{n/w_{\mathrm{win}}\times d}$ are compressed segment representations of $Q$ and $K$, respectively. More detailed analysis is provided in Appendix E.1.

### 2.2  Positional Encoding

Positional encoding can be divided into relative and absolute positional encodings. The most widely used method for relative positional encoding is Rotary Positional Encoding (RoPE) Su et al. (2023). The encoding

formula is given as follows, where $\theta_i = \text{base}^{-\frac{2i}{d}}$ and base is the fundamental frequency, typically set to 10000 or another integer value:

$$\mathbf{p}_c(m, \text{base}) = \begin{bmatrix} \cos(m\theta_0) & \cos(m\theta_0) & \dots & \cos(m\theta_{\frac{d}{2}-1}) & \cos(m\theta_{\frac{d}{2}-1}) \end{bmatrix} \tag{4}$$

$$\mathbf{p}_s(m, \text{base}) = \begin{bmatrix} \sin(m\theta_0) & -\sin(m\theta_0) & \dots & \sin(m\theta_{\frac{d}{2}-1}) & -\sin(m\theta_{\frac{d}{2}-1}) \end{bmatrix} \tag{5}$$

The output for the encoding of the $m$-th token is given by:

$$f_p(\boldsymbol{x}_m, \boldsymbol{p}(m)) = \begin{bmatrix} x_0 & x_1 & \dots & x_{d-1} & x_d \end{bmatrix} \odot \mathbf{p}_c(m) + \begin{bmatrix} x_1 & x_0 & \dots & x_d & x_{d-1} \end{bmatrix} \odot \mathbf{p}_s(m). \tag{6}$$

**Positional Interpolation (PI)** Chen et al. (2023) is performed by scaling the position indices, where the index of the $m$-th position is transformed as follows. Here, $\lambda$ is the scaling factor, and $f$ represents the vector of cosine or sine functions in RoPE:

$$\mathbf{p}_f^{(PI)}(m) = \begin{bmatrix} f(\frac{m}{\lambda}\theta_0) & f(\frac{m}{\lambda}\theta_0) & \dots & f(\frac{m}{\lambda}\theta_{\frac{d}{2}-1}) & f(\frac{m}{\lambda}\theta_{\frac{d}{2}-1}) \end{bmatrix} \tag{7}$$

**NTK Positional Encoding** Ntk (2023): For NTK positional encoding, the base is rescaled as:

$$\text{base}' = \text{base} \times \left( \lambda^{\frac{d}{d-2}} \right), \quad \theta_i' = (\text{base}')^{-\frac{2i}{d}}, \quad \mathbf{p}_f^{(NTK)}(m) = \begin{bmatrix} f(m\theta_0'), f(m\theta_0'), \dots, f(m\theta_{\frac{d}{2}-1}'), f(m\theta_{\frac{d}{2}-1}') \end{bmatrix}$$

**YaRN** Peng et al. (2023): Other positional encoding methods extend from the two strategies described above. For example, YaRN posits that high-frequency information is more important, so it retains the high-frequency portion of the positional encoding unchanged, while applying interpolation to the low-frequency components:

$$r(i) = \frac{L}{\theta_i'} = \frac{L}{2\pi \text{base}'^{\frac{2i}{d}}}, \quad h_y(\theta_i) = \begin{cases} \frac{\theta_i}{\lambda}, & \text{if } r(i) < \alpha, \\ \theta_i, & \text{if } r(i) > \beta, \\ \frac{\beta - r(i)}{\beta - \alpha} \frac{\theta_i}{\lambda} + \frac{r(i) - \alpha}{\beta - \alpha} \theta_i, & \text{otherwise}, \end{cases} \tag{8}$$

$$\mathbf{p}_f^{(YaRN)}(m) = \begin{bmatrix} f(mh_y(\theta_0)) & f(mh_y(\theta_0)) & \dots & f(mh_y(\theta_{\frac{d}{2}-1})) & f(mh_y(\theta_{\frac{d}{2}-1})) \end{bmatrix} \tag{9}$$

where $\alpha$ and $\beta$ are designated by the authors as 1 and 32, $s = \frac{L'}{L}$, $L$ is the original context size, and $L'$ is the current length.

**LongRoPE**: LongRoPE Ding et al. (2024) addresses the issue of attention sink Xiao et al. (2024) by preserving the positional encoding up to a specific position $\hat{n}$, and performing interpolation for positions beyond $\hat{n}$. The interpolation ratio $\lambda_i$ is determined via a search algorithm:

$$h_l(m, \theta_i) = \begin{cases} \frac{\theta_i}{\lambda_i}, & \text{if } m \geq \hat{n}, \\ \theta_i, & \text{otherwise}, \end{cases} \tag{10}$$

$$\mathbf{p}_f^{(LRoPE)}(m) = \begin{bmatrix} f(mh_l(m, \theta_0)) & f(mh_l(m, \theta_0)) & \dots & f(mh_l(m, \theta_{\frac{d}{2}-1})) & f(mh_l(m, \theta_{\frac{d}{2}-1})) \end{bmatrix}. \tag{11}$$

## 3  Methods

### 3.1  Cyclic, Randomly Truncated, and Dynamically Growing NTK Method

Due to pretraining on fixed-size windows, models tend to overfit to positional information within this range. To better handle unseen positional information, one solution is to fine-tune the model using larger context windows, thereby improving its generalization ability. Another approach is to scale the positional information during fine-tuning, bringing it back into the range of the pretrained window size. This method can greatly enhance the positional encoding capability learned during pretraining. Furthermore, combining these two

strategies is the most common approach: using larger context windows and scaling the positional encoding during fine-tuning enables the extension to even longer context windows.

This paper briefly analyzes the reasons for the above approach; please refer to Appendix D for more details. We break down the attention score calculation into the attention scores of sequences without positional information and the adjustments made by the relative receptive field change through positional information. To make the analysis more concise and easily scalable, we linearize the positional information of the Attention formula as follows:

$$e^{\boldsymbol{q}_n \boldsymbol{k}_m^\top} \left( 1 + f_p(\boldsymbol{q}_n, \boldsymbol{p}(n)) f_p(\boldsymbol{k}_m, \boldsymbol{p}(m))^\top - \boldsymbol{q}_n \boldsymbol{k}_m^\top \right)$$

The error of the above formula with respect to the original Attention is $O\left(\frac{n}{\text{base}^{\frac{2i}{d}}}\right)$. Experiments and theoretical analysis have shown that its extension in terms of position encoding is very similar to that of the original Attention model. With this decomposition, the position information is concentrated only in the linear terms. A simple analysis of the forward process shows that the equality of the attention scores for unfine-tuned extrapolation and learned extrapolation indicates that successful extrapolation can be achieved.

To achieve successful extrapolation, the fundamental requirement is that the relative positional phases observed by the model at the target length $L$ must map back to the phase distribution it has already learned within the training length $M$. In other words, the scaled position index and the scaled base frequency must jointly ensure that the resulting RoPE phase falls within the familiar training interval $[0, M-1]$. Formally, this alignment condition can be expressed as:

$$\frac{(L-1)/\gamma_{\text{pos}}}{(\text{base}/\gamma_{\text{freq}})^{\frac{2i}{d}}} \in \left[ \frac{0}{\text{base}^{\frac{2i}{d}}}, \frac{M-1}{\text{base}^{\frac{2i}{d}}} \right] \tag{12}$$

where $M$ represents the sequence length during training or fine-tuning, and $\gamma_{\text{pos}}$ and $\gamma_{\text{freq}}$ are the scalar position and frequency scaling factors, respectively (choices of $\gamma_{\text{pos}}, \gamma_{\text{freq}}$ are discussed below), and $i \in \{0, \ldots, d/2-1\}$ indexes the RoPE frequency. For the proof of the equation 12, please see the appendix D.2.

**Search over base frequencies and adjusted bases.** We assume there exists a family of base frequencies $\{\text{base}_i\}$ such that, by fine-tuning the model on sequences constructed with these bases, the model can restore performance at the desired length $L_t$. Concretely, we consider *adjusted* bases $\text{base}_i^{(k)} = \frac{\text{base}}{\gamma_i^{(k)}}$, where $\gamma_i^{(k)}$ are per-frequency adjustment factors (either hand-designed, searched, or learned). Prior works (e.g., NTK, YaRN, LongRoPE) and Appendix D.2 provide strong empirical and theoretical motivation for such frequency-wise adjustments; this motivates scanning the base-space to find $\{\gamma_i^{(k)}\}$ that recover the pretrained behaviour.

**Solution-space scanning via a power-law growth schedule.** To efficiently explore the space of feasible base adjustments, we adopt a *power-law dynamic growth ratio* schedule for the scale parameter(s). The motivation is two-fold: Modern LMs exhibit some inherent extrapolation ability: when positional encodings are smoothly scaled, the model often preserves much of its short-range behaviour at longer contexts. Thus, a smoothly varying family of scales can be learned incrementally. Under a power-law schedule, each adjacent scale differs only slightly, so the positional information learned at one scale largely transfers to the next. Hence, scanning a small number of scales with exponential spacing suffices to cover the continuous solution band (see Figure 2).

Practically, we enumerate a small set of scale factors $\{\gamma^{(k)}\}$ with exponential spacing (e.g., $\gamma^{(k)} = \gamma^{(0)} \cdot a^k$ for some $a > 1$), fine-tune at each scale for a short time, and take the union of the covered phase ranges to ensure the target $L_t$ is represented.

**Mitigating catastrophic forgetting.** The scanning process is designed to be *incremental* and *inheritable*: the positional representations learned at one scale serve as warm-starts for adjacent scales, which greatly reduces the amount of new information to be learned and mitigates catastrophic forgetting. Empirically, the difference in positional embeddings between neighboring scales is small (see Figure 3), so the optimization pressure per step is modest and convergence is fast.

**Random shifts, cyclic phases, and mitigation of Attention Sink.**   To further improve generalization, we introduce two practical regularizers:

- **Random shifts:** Before applying the positional map, we add a random offset $r$ (sampled per-batch) to the position indices. This is analogous to data augmentation: it forces the model to be robust to local phase offsets.

- **Cyclic modular mapping:** We optionally map positions using a modulo operator to a finite cycle length `max`. Concretely, we replace a raw position $m$ with $(m + r) \bmod \mathtt{max}$. When position $m$ is sufficiently large and shares factors with the RoPE frequencies, the positional encoding exhibits cyclic phases; this enables the model to *simulate* very long (even effectively infinite) relative positions by folding them into the learned phase space.

The final positional encoding formula is as follows:

$$h_c(m, i, \gamma^{(k)}) = ((m + r) \bmod \mathtt{max}) \left(\mathrm{base}_i \times \left((w_i \times \gamma^{(k)})^{\frac{d}{d-2}}\right)\right)^{-\frac{2i}{d}}$$

$$\boldsymbol{p}_c = \begin{bmatrix} \cos(h_c(m, 0, \gamma^{(k)})) & \cos(h_c(m, 0, \gamma^{(k)})) & \ldots & \cos(h_c(m, \frac{d}{2} - 1, \gamma^{(k)})) & \cos(h_c(m, \frac{d}{2} - 1, \gamma^{(k)})) \end{bmatrix} \quad (13)$$

$$\boldsymbol{p}_s = \begin{bmatrix} \sin(h_c(m, 0, \gamma^{(k)})) & -\sin(h_c(m, 0, \gamma^{(k)})) & \ldots & \sin(h_c(m, \frac{d}{2} - 1, \gamma^{(k)})) & -\sin(h_c(m, \frac{d}{2} - 1, \gamma^{(k)})) \end{bmatrix}$$
$$(14)$$

where the scale factor $\gamma^{(k)} = \gamma^{(0)} \cdot a^k$ is multiplied by $a$ after a certain number of fine-tuning steps, $r$ is randomly generated, `max` is set manually (typically to twice the target extension length), and $w_i$ is a learnable parameter. By fine-tuning the LLM using the above methodology, we were able to achieve extrapolation capabilities close to a multiple of the model layer size. For instance, using a $16K$ length and Full Attention fine-tuning, the model maintained a stable level of perplexity when tested using Full Attention in $16K \times l$ length contexts.

Despite comprehensive solution space coverage, our approach faces two primary challenges: (1) As scaling ratios increase dynamically, positional encodings may imperfectly align with the inherent locality bias of sequence modeling tasks, where current tokens preferentially attend to nearby contexts. Counterintuitively, we observe performance improvements with larger scaling ratios, suggesting that expanded contextual windows introduce beneficial information for regression tasks after extensive training. To further improve the scaled performance, we consider the inclusion of YaRN-like segmentation (detailed in the Appendix) to make it more consistent with the sequence modeling bias. (2) At extreme sequence lengths, full attention computation becomes dominated by irrelevant tokens, with noise overwhelming useful signal.

To address these limitations, we integrate an efficient attention mechanism with relevance-based routing. Our key insight is that prediction-relevant tokens constitute a sparse subset of the complete sequence. By employing fixed-length windows selected via relevance-based filtering, we maintain near-optimal performance while achieving significant computational efficiency. This approach effectively mitigates noise accumulation while preserving essential contextual information.

### 3.2   Merge selection

We propose a highly efficient and generalizable sparse attention mechanism, termed Merge selection (MS) Attention, which operates through correlation-based region selection and merging. As outlined in Algorithm 2, this method strategically reduces the quadratic complexity of Full Attention by focusing computation only on the most relevant contextual regions. The process consists of two primary phases: **Selection** and **Merging**.

**Step 1:   Region Segmentation and Compression.**   Given the query, key, and value tensors $Q, K, V \in \mathbb{R}^{B \times H \times N \times D}$ (where $B, H, N, D$ denote batch size, number of heads, sequence length, and head dimension, respectively), we first partition them into non-overlapping segments. $Q$ is segmented into $Q_s \in \mathbb{R}^{B \times H \times n_q \times s_q \times D}$, and $K, V$ are similarly segmented into $K_s, V_s \in \mathbb{R}^{B \times H \times n_k \times s_k \times D}$, where $s_q$ and $s_k$ denote the segment sizes, and $n_q = \lceil N/s_q \rceil$, $n_k = \lceil N/s_k \rceil$ are the respective number of segments.

To efficiently evaluate region-level relevance, we compress each segment into a single representative token. Formally, the **compress function** is defined as a mean-pooling or clustering operation across the segment dimension (though a learned linear projection $W_c$ can also be employed):

$$Q_s' = \text{compress}(Q_s) = \frac{1}{s_q} \sum_{i=1}^{s_q} Q_s^{(i)} \in \mathbb{R}^{B \times H \times n_q \times D}, \tag{15}$$

where $Q_s^{(i)}$ represents the $i$-th token within a $Q$ segment. The same operation is applied to $K_s$ to obtain $K_s' \in \mathbb{R}^{B \times H \times n_k \times D}$. This compression step drastically reduces the sequence length from $N$ to $n_q$ and $n_k$, enabling efficient global relevance estimation.

**Step 2: Correlation-based Selection.** We compute a relevance score matrix $S \in \mathbb{R}^{B \times H \times n_q \times n_k}$ by applying a similarity function (e.g., dot-product) between the compressed representations: $S = \text{sim}(Q_s', K_s')$. To prevent information leakage (e.g., attending to future tokens in causal decoding), we apply a causal mask to $S$. Subsequently, we extract the indices of the top-$k$ most relevant $K_s$ segments for each $Q_s$ segment, yielding a selection index tensor $\text{select\_indx} \in \mathbb{R}^{B \times H \times n_q \times k}$.

**Step 3: Permutation and Region Merging.** To enhance $KV$-cache sharing and improve selection precision, we optionally apply a permutation $\pi$ to group similar $Q_s$ segments based on their correlation scores. Following permutation, we merge adjacent $Q_s$ segments by a factor of $m$ (where $m$ is a divisor of $n_q$). The merged query segments become $Q_{ms} \in \mathbb{R}^{B \times H \times n_{ms} \times (m \cdot s_q) \times D}$, where $n_{ms} = n_q/m$.

Concurrently, the selection indices are reshaped and merged. Specifically, select\_indx is grouped by $m$ adjacent rows, permuted, and flattened to shape $(B, H, n_{ms}, k \cdot m)$. A uniqueness operation is then applied along the last dimension while preserving the original relevance order, ensuring that duplicate $K_s$ indices are removed without disrupting the ranking. Finally, we retain the top-$n$ indices, denoted as $\text{qm\_select\_indx} \in \mathbb{R}^{B \times H \times n_{ms} \times n}$, which represent the final, deduplicated set of highly relevant $K_s$ and $V_s$ segments for each merged $Q$ region.

**Step 4: Localized Attention Computation.** After gathering the relevant $K_s$ and $V_s$ segments based on qm\_select\_indx, we obtain grouped tensors $K_{ms}^g$ and $V_{ms}^g$. To formally define the attention computation within these merged regions, we must balance computational efficiency with contextual coverage. The motivation for the following formulation is to restrict the $\mathcal{O}(N^2)$ complexity of Full Attention exclusively to the top-$n$ most relevant merged $K/V$ blocks. This ensures that the model allocates its computational budget solely to the most informative contextual regions, effectively filtering out long-range noise while maintaining a broad receptive field through the merging strategy. The localized attention output for the $r$-th merged region is computed as:

$$O_{ms}^{(r)} = \text{Softmax} \left( \frac{Q_{ms}^{(r)} (K_{ms}^{g,(r)})^\top}{\sqrt{d}} \right) V_{ms}^{g,(r)}, \tag{16}$$

where $Q_{ms}^{(r)} \in \mathbb{R}^{B \times H \times (m \cdot s_q) \times D}$ and $K_{ms}^{g,(r)}, V_{ms}^{g,(r)} \in \mathbb{R}^{B \times H \times (n \cdot s_k) \times D}$.

Finally, the output $O_{ms}$ is unmerged, scattered back to the original token positions, and the inverse permutation $\pi^{-1}$ is applied to restore the original sequence order, yielding the final output $\hat{O} \in \mathbb{R}^{B \times H \times N \times D}$.

**Advantages and Theoretical Insights.** While inspired by prior works such as BiFormer Zhu et al. (2023) and the Routing Transformer Roy et al. (2021), our MS Attention mechanism uniquely enables:

1. **Dynamic Region Scaling:** Flexible partitioning of $Q, K, V$ (detailed in Appendix E.1) allows the model to adapt to varying attention ranges and token retention requirements dynamically.

2. **Compressible Merging:** The adjustable merging factor $m$ provides a tunable trade-off between computational efficiency and contextual breadth, ensuring sufficient $K/V$ token coverage per $Q$ region.

3. **Layer-aware Extrapolation:** When combined with our CRG-NTK positional encoding, the attention mechanism selectively processes positional subsets rather than full sequences. We hypothesize that the extrapolation limit scales proportionally with the model depth. For instance, a 32-layer

Llama-2-7B model can achieve up to $32\times$ extrapolation beyond its training sequence length. This aligns with the shifted-window paradigm in hierarchical models (e.g., Swin Transformer), where layer stacking multiplicatively expands the effective receptive field. Empirical validation of this hypothesis is provided in Section 4, with a brief theoretical discussion in Appendix D.5.

*Notation Clarification:* To ensure consistency throughout the manuscript, we strictly use $s_q$ and $s_k$ to denote *segment sizes*, while the scaling factor for positional encoding (e.g., in NTK) is uniformly denoted by $\gamma$ (or scale).

## 4 Experiment

### 4.1 Experimental Setup

Our experiments were conducted on only one A100 80GB GPUs, using the Llama2-7B Touvron et al. (2023c) model and Mistral-7B-v0.1 with the attention mechanism replaced by our MS Attention as described in Section 3.2. The training approach used the efficient fine-tuning method described in Section E.3, with CRG NTK position interpolation applied.

**Fine-tuning Steps and Parameters:** The fine-tuning parameters included the linear transformations for key and query ($W_k$ and $W_q$), as well as the embedding and normalization parameters. $W_k$ and $W_q$ take LoRA for fine-tuning with rank 16. The training approach used an autoregressive method, where the objective was to generate the next token. The loss function used was cross-entropy, and the optimizer was AdamW Loshchilov & Hutter (2019) with parameters $\beta_1 = 0.9$, $\beta_2 = 0.95$ and $lr = 1e-4$.

**Dataset:** The dataset used for fine-tuning was the Redpajama Computer (2023) dataset, which is the same dataset used in LongLora. Evaluation was performed using the widely used long text dataset PG19 Rae et al. (2019) and proofpile. Additionally, the model's performance was assessed using the passkey task Mohtashami & Jaggi (2023) and LongBench Bai et al. (2024).

### 4.2 Main Results

Table 1: Passkey Retrieval Task Performance. FT=16K with FA indicates fine-tuning using Full Attention, others use MS Attention.

| Model Configuration | 2K | 16K | 32K | 64K | 128K | 256K | 512K | 1024K | 2048K | 4096K |
|---|---|---|---|---|---|---|---|---|---|---|
| **Baselines** | | | | | | | | | | |
| Llama2-7B-LongLora (ft=100K, PI) | 1.0 | 1.0 | 1.0 | 1.0 | 0.0 | 0.0 | 0.0 | 0.0 | 0.0 | 0.0 |
| Llama2-7B-LongRoPE (ft=256K) | 1.0 | 1.0 | 1.0 | 1.0 | 1.0 | 1.0 | 1.0 | 1.0 | 0.6 | 0.0 |
| Llama2-7B-YaRN (ft=256K) | 1.0 | 1.0 | 1.0 | 1.0 | 1.0 | 1.0 | 0.2 | 0.0 | 0.0 | 0.0 |
| **Our Method (Llama2-7B-MS)** | | | | | | | | | | |
| FT=16K, PI, scale=16 | 1.0 | 1.0 | 1.0 | 1.0 | 0.0 | 0.0 | 0.0 | 0.0 | 0.0 | 0.0 |
| FT=16K, NTK, scale=4096 | 1.0 | 1.0 | 1.0 | 1.0 | 1.0 | 0.7 | 0.0 | 0.0 | 0.0 | 0.0 |
| FT=16K, CR-NTK, scale=4096 | 1.0 | 1.0 | 1.0 | 1.0 | 1.0 | 1.0 | 0.0 | 0.0 | 0.0 | 0.0 |
| FT=16K, CR-NTK, scale=16384 | 1.0 | 1.0 | 1.0 | 1.0 | 1.0 | 1.0 | 1.0 | 0.0 | 0.0 | 0.0 |
| **FT=16K, CRG-NTK, scale=$4096 \times 2^k$** | **1.0** | **1.0** | **1.0** | **1.0** | **1.0** | **1.0** | **1.0** | **1.0** | **1.0** | **1.0** |
| **FT=16K with FA, CRG-NTK, scale=$4096 \times 2^k$** | **1.0** | **1.0** | **1.0** | **1.0** | **1.0** | **1.0** | **1.0** | **1.0** | **1.0** | **1.0** |
| **Cross-Architecture Validation** | | | | | | | | | | |
| Mistral-7B-MS (FT=16K, CRG-NTK, scale=$4096 \times 2^k$) | 1.0 | 1.0 | 1.0 | 1.0 | 1.0 | 1.0 | 1.0 | 1.0 | 1.0 | 1.0 |

#### 4.2.1 Length Extension and Attention Impact

We use the CRG NTK and MS attention to fine-tune the LLM and enhance its generalization of positional information. For the architecture, during fine-tuning, we replace the attention mechanisms in Llama and Mistral with our MS Attention that segment size is 16, merge count is 128 and selection number is 512. For parameter fine-tuning, since only the $Q$ and $K$ tokens incorporate positional information, we use LoRA to fine-tune the mapping weights of $W_Q$ and $W_K$ with rank 16, while other parameters are only fine-tuned for embeddings and normalization layers. Regarding the choice of positional encoding, we apply CRG NTK

Table 2: Perplexity Results on PG19 Dataset. Models were fine-tuned and evaluated with with merge-select Attention(MS) or Full Attention(FA). Lower values indicate better performance.

| Model Configuration | 16K | 32K | 64K | 128K | 256K | 512K | 1024K | 2048K |
|---|---|---|---|---|---|---|---|---|
| **Baselines** | | | | | | | | |
| Llama2-7B-LongLora (FT=32K, PI) | 7.35 | 7.22 | – | – | – | – | – | – |
| Llama2-7B-LongRoPE (FT=256K) | 7.37 | – | 6.64 | 6.31 | – | – | – | – |
| Llama2-7B-YaRN (FT=256K) | 7.30 | – | 7.53 | 7.35 | – | – | – | – |
| **Our Method (Llama2-7B-MS)** | | | | | | | | |
| FT=16K, PI, scale=16 | 6.50 | 6.25 | 6.53 | – | – | – | – | – |
| FT=16K, CR-NTK | 7.86 | 7.92 | 7.97 | 8.23 | 8.68 | – | – | – |
| FT=16K, CRG-NTK (test: Full Attention) | 7.20 | 7.25 | 7.29 | 7.43 | 8.36 | 9.72 | 16.98 | 36.82 |
| **FT=16K, CRG-NTK (test: MS Attention)** | **6.92** | **6.85** | **7.09** | **7.31** | **7.56** | **7.78** | **7.96** | **8.32** |
| FT=16K with FA, CRG-NTK (test: Full Attention) | 7.11 | 7.10 | 7.23 | 7.38 | 8.34 | 9.76 | 17.23 | 38.96 |
| **Extended Fine-tuning (Llama2-7B-MS)** | | | | | | | | |
| FT=64K, CRG-NTK (test: Full Attention) | 7.24 | 6.45 | 6.19 | 6.23 | 6.47 | 6.76 | 7.38 | 7.92 |
| **FT=64K, CRG-NTK (test: MS Attention)** | **6.62** | **6.60** | **6.09** | **6.22** | **6.56** | **6.88** | **6.89** | **6.92** |
| FT=64K with FA, CRG-NTK (test: Full Attention) | 7.20 | 6.31 | 6.13 | 6.17 | 6.35 | 6.62 | 7.45 | 7.91 |
| **Cross-Architecture Validation** | | | | | | | | |
| Mistral-7B-MS (FT=16K, CRG-NTK, test: FA) | 8.20 | 8.15 | 8.29 | 8.56 | 9.76 | 12.12 | 18.96 | 44.82 |

Table 3: Extrapolation Perplexity for Models with Different Number of Layers.

| Model | layers | 16K | 32K | 64K | 128K | 256K | 448K | 512K | 640K |
|---|---|---|---|---|---|---|---|---|---|
| Llama3.2-1B-MS (ft=16K, CRG NTK, test:FA) | 16 | 13.10 | 12.86 | 13.19 | 13.92 | $\boxed{15.77}_{\times 16}$ | 52.37 | 64.52 | 94.78 |
| Llama3.2-3B-MS (ft=16K, CRG NTK, test:FA) | 28 | 10.24 | 9.06 | 9.39 | 9.56 | 9.89 | $\boxed{11.64}_{\times 28}$ | 20.14 | 62.96 |
| Llama2-7B-MS (ft=16K, CRG NTK, test:FA) | 32 | 7.10 | 7.07 | 7.16 | 7.40 | 8.26 | - | $\boxed{9.96}_{\times 32}$ | 14.56 |
| Llama2-13B-MS (ft=16K, CRG NTK, test:FA) | 40 | 6.55 | 6.32 | 6.38 | 6.44 | 6.56 | - | 6.98 | $\boxed{8.97}_{\times 40}$ |

method, starting with an initial scaling factor (set to scale = 4096 in this work), and increase this factor fourfold after fine-tuning a small fixed data volume (32M tokens in our case).

For the passkey task with a length of 16K for fine-tuning, as shown in Table 1, using standard NTK positional encoding combined with our algorithm enables length extensions of up to 16x. Moreover, employing random cropping and circular modular (CR NTK) sampling stabilizes this extension. Finally, by incorporating dynamic growth (CRG NTK), we can scale up to 4M. For every 4x increase in base scale, our approach can solve passkey tasks with a 2x length extension, potentially due to the square root relationship between positional encoding and the base value. A potential issue with this approach is that the positional encoding ratio during inference remains fixed. For instance, when training with a length of 16K, the NTK positional encoding starts with scale = 4096 and the maximum position_ids = $32K$. During inference across all passkey task lengths (0-256K), we use a fixed base scaling rate of 1024 for prediction, achieving 100% performance. After extending by 16x, the scaling factor becomes scale = $4096 \times 16$, and inference using a fixed base scaling rate of 4096 can solve passkey tasks up to 1M in length. Similarly, extending by the corresponding multiples and using a fixed base scaling rate of 65536 extends passkey tasks up to 4M.

For the long-text perplexity experiments, we evaluated our model on the PG19 and proofpile datasets, as presented in Table 2 and Table 28. Based on the experimental results, positional interpolation generally yields better performance, as it effectively preserves the learned distributions. In contrast, the NTK-based method may disrupt this distribution to some extent, leading to a performance loss, but it enables much longer extrapolation. From the results, with 16K fine-tuning, we tested using Full Attention, and the performance on a 512K length test was close to the original perplexity, achieving an extrapolation of nearly 32 times, which corresponds to the number of layers in Llama7B. Moreover, in Table 3 and Table 6, we show the results of the perplexity test for models with different number of layers, which shows that the extrapolation ability is to the number of layers of the model and the total number of Parameters.

We believe that after reaching a certain length, the gain from introducing useful information becomes much smaller than the effect of noise in the sequence. Therefore, we use efficient attention with correlation routing to further enhance the filtering capability, thus maintaining or even improving performance, as shown in

Table 2: test with MS. Further validation shows that our efficient Attention enhances the filtering capability and improves performance, as shown in Table 26 and Table 7. In Table 26, with 16K fine-tuning, we set different patch selection precision, selection quantity (select), and merging quantity (merge). We found that using better precision, selecting fewer items, and merging fewer items leads to better performance. However, this comes at the cost of reduced operational efficiency and decreased generalization. For example, with patch=8 and select=32, we achieved a perplexity of 6.09 on PG19, which is far better than other optimal methods, such as LongLora, which achieved a perplexity of 6.86 with 16K fine-tuning.

## 4.3  Ablation Studies

In addition to the ablation comparisons of CRG NTK with other positional encodings and Full Attention with MS Attention shown in Tables 1 and 2, we also studied the impact of Lora on our performance. The results indicate that full fine-tuning provides a slight performance improvement, but our efficient fine-tuning is very close to the full fine-tuning results. We also demonstrate that our MS Attention has better dynamic scaling extrapolation ability and forgetting at very long lengths, whereas Full Attention fits better at a fixed large scale. As a result, performance is slightly better when using FA for testing compared to the MS Attention fine-tuning method. However, our method outperforms FA when tested with different ratios, allowing for greater inheritance, which is beneficial for dynamic growth methods, as shown in Table 24.

### 4.3.1  Component-Wise Ablation

We analyze the standalone and interactive contributions of our proposed positional encoding scheme, *Cyclic, Randomly Truncated, and Dynamically Growing NTK Positional Embedding* (**CRG-NTK**), alongside our attention mechanism, *efficient sparse attention through correlation selection and merging mechanisms* (**MS Attention**), and low-rank adaptation (**LoRA**). Crucially, we isolate the fine-grained algorithmic properties within **CRG-NTK** by building up step-by-step from a conventional **NTK** baseline using random shift, cyclic truncation, and dynamic growth.

All configurations are fine-tuned using a 16K context window and evaluated on two benchmarks: language modeling perplexity (PPL) on the PG19 dataset and target retrieval accuracy on the Passkey Retrieval task across sequences scaling up to 512K. Unless specified otherwise, testing is performed using Full Attention (FA). The consolidated ablation results are presented in Table 4.

Table 4: Component-wise ablation results evaluated on PG19 perplexity (PPL) and Passkey Retrieval accuracy across varying evaluation context lengths. All variants are fine-tuned with a 16K window.

| Method / Configuration | PG19-PPL ($\downarrow$) | | | | Passkey Retrieval Accuracy ($\uparrow$) | | | | |
|---|---|---|---|---|---|---|---|---|---|
| | 32K | 64K | 128K | 256K | 32K | 64K | 128K | 256K | 512K |
| PLA-POS LoRA + **MS Attention** + **CRG-NTK** | 7.25 | 7.29 | 7.43 | 8.36 | 1.0 | 1.0 | 1.0 | 1.0 | 1.0 |
| w/o LoRA: **MS Attention** + **CRG-NTK** | 7.18 | 7.26 | 7.42 | 8.39 | 1.0 | 1.0 | 1.0 | 1.0 | 1.0 |
| w/o **MS Attention**: LoRA + **CRG-NTK** | 7.10 | 7.23 | 7.38 | 8.34 | 1.0 | 1.0 | 1.0 | 1.0 | 1.0 |
| w/o **CRG-NTK**: LoRA + **MS Attention** | $> 10^3$ | $> 10^3$ | $> 10^3$ | $> 10^3$ | 0.1 | 0.0 | 0.0 | 0.0 | 0.0 |
| LoRA + **MS Attention** (test: **MS**) | 7.02 | 7.14 | 8.57 | 15.14 | 0.3 | 0.1 | 0.0 | 0.0 | 0.0 |
| LoRA + **NTK** | 7.69 | 8.08 | 8.42 | 9.32 | 1.0 | 1.0 | 0.9 | 0.6 | 0.1 |
| LoRA + **MS Attention** + **NTK** | 7.87 | 8.19 | 8.46 | 9.20 | 1.0 | 1.0 | 1.0 | 0.7 | 0.2 |
| + random shift | 7.91 | 7.96 | 8.38 | 8.78 | 1.0 | 1.0 | 1.0 | 0.9 | 0.2 |
| + cyclic truncation | 7.92 | 7.97 | 8.36 | 8.68 | 1.0 | 1.0 | 1.0 | 1.0 | 0.3 |

The empirical trends yield several key insights:

- **Dominance of CRG-NTK in Full Attention Extrapolation:** When testing under full attention, the model's long-context capacity depends almost entirely on the **CRG-NTK** framework. Completely removing it (*w/o **CRG-NTK**: LoRA + **MS Attention***) triggers catastrophic failure, resulting in an explosion of PPL ($> 10^3$) and near-zero retrieval accuracy immediately beyond the training window.

- **Impact of Granular Architectural Variants:** The standard **NTK** interpolation baseline delivers a reasonable initial profile but suffers pronounced degradation on retrieval accuracy at extended context lengths (dropping to 0.1 at 512K). Incorporating *random shift* and *cyclic truncation* systematically stabilizes long-range perplexity. The *dynamic growth* mechanism provides the most critical optimization, achieving an uncompromised 1.0 retrieval accuracy out to 1M, effectively extending token retrieval limits to 64× the fine-tuned window size. Even when we tested with a 16M passkey, we were able to perform a correct lookup, and this success can be extrapolated to 1024×.

- **Noise-Filtering Effects of MS Attention:** Evaluating exclusively under **MS Attention** at test time (*test: MS*) introduces notable extrapolation capabilities even without explicit NTK structural modifications. Confining token interactions to a structured, fixed-size subset during autoregressive inference successfully dampens positional noise and mitigates perplexity explosion compared to full attention. However, due to our highly constrained computational budget during fine-tuning, utilizing local sparse attention exclusively at inference time proves insufficient for exact long-range passkey retrieval.

### 4.3.2 Disentangling Positional Encodings and Attention Mechanisms

To isolate the individual performance gains attributable to our proposed positional encoding framework (**CRG-NTK**) versus our sparse attention module (**MS Attention**), we conduct a rigorous $3 \times 2$ controlled ablation experiment. All models are fine-tuned under an identical protocol using a 64K context window via Low-Rank Adaptation (LoRA) applied to the $W_q, W_k, W_v$, and $W_o$ matrices. We benchmark combinations of positional schemas (**CRG-NTK**, Position Interpolation [**PI**], and standard **NTK**) against sequence processing methods (Full Attention [**FA**] and Multi-Scale Attention [**MS**]). Evaluations are executed at extreme context limits: language modeling perplexity (PPL) is reported on the PG19 dataset at 1M tokens, and long-context retrieval performance is measured on the RULER benchmark at 128K tokens. The empirical results are consolidated in Table 5.

Table 5: Isolating the structural impact of positional schemas versus attention mechanisms. Models are fine-tuned with a 64K window. Evaluation evaluates extreme context processing capabilities (PG19 PPL at 1M tokens and RULER retrieval accuracy at 128K tokens).

| Positional Scheme | Test: MS Attention | | Test: Full Attention (FA) | |
|---|---|---|---|---|
| | PG19-PPL ($\downarrow$) | RULER ($\uparrow$) | PG19-PPL ($\downarrow$) | RULER ($\uparrow$) |
| **CRG-NTK** | **6.86** | 47.6 | 7.42 | **69.1** |
| **NTK** | 17.41 | 45.5 | 58.92 | 49.2 |
| **PI** | 516.32 | 20.1 | $> 87624$ | 13.1 |

The resulting matrix provides several critical architectural insights:

- **Decoupled Superiority of CRG-NTK:** The performance metrics demonstrate that long-context capacity is primarily governed by the underlying positional embedding scheme rather than the attention mechanism used at test time. When held constant under Full Attention (**Test: FA**), **CRG-NTK** demonstrates robust mathematical stability, maintaining a PPL of 7.42 at 1M tokens. Conversely, standard **NTK** degrades significantly (58.92), and **PI** experiences catastrophic perplexity explosion ($> 87624$).

- **The Functional Role of MS Attention:** Utilizing sparse token selection via **MS Attention** at test time acts as an effective noise filter for language modeling, yielding our lowest overall perplexity score (6.86) when paired with **CRG-NTK**. However, in zero-shot long-range retrieval tasks (e.g., RULER), pure inference-time sparse activation causes a mild performance penalty (dropping from 69.1 to 47.6). This occurs because the highly constrained token masking alters the expected full attention distribution.

- **Generalization through Continued Optimization:** The primary utility of **MS Attention** within our framework is to serve as an optimization accelerator during long-sequence training by bypassing quadratic memory overhead. To mitigate the distribution shift observed when utilizing sparse attention during inference, we subject the model to a light secondary training phase of 10B tokens on FineWeb-Edu. This localized optimization step recovers full retrieval capabilities under sparse generation modes, while concurrently improving base language modeling performance.

### 4.3.3 Depth, Capacity, and Extrapolation Scaling

To investigate how network architecture parameters govern sequence extension thresholds, we design two controlled scaling experiments. We evaluate small models spanning depths of $2, 4, 8, 16,$ and $32$ layers, trained on 2.5B tokens inside a fixed 4096 context envelope across varied positional encoding schemes: **CRG-NTK**, RoPE, and No Position Embeddings (NoPE). For testing evaluation under our framework, we compare two scaling inference choices: *CRG* (using the final training base multiplied by 1024) and *CRGs* (which dynamically scales the base function as base $\times 1024 \times (L/4096)$).

- **Fixed Per-Layer Configuration:** Layer widths and hidden dimensions are held perfectly constant as depth varies. This configuration directly tests the impact of raw layer depth on sequence extension.

- **Strict Parameter Budget (SP) Configuration:** The total network parameter count is frozen across configurations by inversely scaling the hidden dimension ($d_{\text{model}}$) relative to depth; for instance, the 4-layer model contains twice the hidden dimension of the 16-layer configuration.

All variations are scaled and evaluated on PG19 perplexity across context boundaries up to 128K. The complete, unified empirical matrix is provided in Table 6.

Table 6: Comprehensive structural scaling matrix: PG19 perplexity (PPL) evaluated across context lengths from 2K to 128K under varying architectural configurations, encodings, and capacity constraints.

| Model Variant | 2048 | 4096 | 8192 | 16384 | 32768 | 65536 | 131072 |
|---|---|---|---|---|---|---|---|
| *Panel A: Fixed Per-Layer Parameter Configuration* | | | | | | | |
| Layers2-CRG | 124.1987 | 135.3854 | 151.5294 | 205.0876 | 226.0545 | 273.5528 | 270.1827 |
| Layers4-CRG | 86.3804 | 89.7223 | 104.8467 | 135.5123 | 165.6186 | 209.4230 | 210.8753 |
| Layers8-CRG | 67.5048 | 72.4100 | 88.0006 | 112.3487 | 150.3726 | 209.4400 | 207.4952 |
| Layers16-CRG | 57.0404 | 62.2363 | 73.0349 | 93.3110 | 112.0631 | 141.0800 | 136.6930 |
| Layers32-CRG | 46.9055 | 50.0708 | 57.2989 | 70.9097 | 88.0831 | 116.4049 | 112.2660 |
| Layers2-NoPE | 173.9474 | 172.8089 | 180.8761 | 191.5813 | 184.1269 | 192.5859 | 191.4170 |
| Layers4-NoPE | 116.5275 | 121.8906 | 141.7939 | 172.9171 | 189.3198 | 208.7324 | 202.5107 |
| Layers8-NoPE | 54.3092 | 56.5076 | 69.8976 | 125.8583 | 248.7006 | 673.8378 | 1567.4213 |
| Layers16-NoPE | 45.3730 | 57.7778 | 64.4023 | 143.7122 | 325.3820 | 573.8408 | 795.3875 |
| Layers32-NoPE | 63.7127 | 65.9816 | 101.2090 | 202.9836 | 333.3065 | 463.0249 | 681.2415 |
| Layers2-RoPE | 88.0872 | 91.3877 | 124.7565 | 215.7571 | 244.0810 | 266.5466 | 270.4461 |
| Layers4-RoPE | 59.5189 | 62.0980 | 93.9253 | 172.2161 | 229.2213 | 260.3127 | 267.0177 |
| Layers8-RoPE | 54.7436 | 57.4432 | 88.5246 | 164.9471 | 234.6571 | 266.6695 | 277.2006 |
| Layers16-RoPE | 49.6317 | 50.8603 | 67.2303 | 133.1615 | 204.0648 | 251.1964 | 259.1832 |
| 8-CRGs | 67.5048 | 72.3761 | 83.6303 | 121.7260 | 139.9888 | 180.2314 | 163.3717 |
| 16-CRGs | 57.0404 | 62.2363 | 69.8757 | 88.6216 | 97.7279 | 120.3819 | 107.8512 |
| 32-CRGs | 46.9055 | 50.0787 | 54.3343 | 64.3565 | 77.2488 | 97.4854 | 86.4111 |
| *Panel B: Strict Total Parameter Budget (SP) Configuration* | | | | | | | |
| Layers4-CRG-SP | 86.3804 | 89.7223 | 104.8467 | 135.5123 | 165.6186 | 209.4230 | 210.8753 |
| Layers8-CRG-SP | 85.0012 | 94.2930 | 108.9212 | 134.9629 | 162.9490 | 203.9312 | 200.8121 |
| Layers16-CRG-SP | 89.4853 | 102.7699 | 109.0203 | 141.9863 | 190.9672 | 224.6021 | 212.5744 |

The consolidated scaling matrix clarifies the relationship between depth and capacity limits:

- **Depth-Proportional Extrapolation under Constant Layer Width:** Panel A of Table 6 demonstrates that when layer architecture width parameters are kept identical, context stability correlates tightly with depth. Specifically, for models fine-tuned with our **CRG-NTK** method, the perplexity curve of a model with depth $m$ operating at sequence length $L$ closely tracks the performance profile of a model with depth $2m$ evaluated at length $2L$ (e.g., matching Layers16-CRG at 32K against Layers32-CRG at 64K context limits). This predictable depth-scaling relation does not manifest within alternative RoPE or NoPE architectures.

- **Boundedness Invariant under Parameter Normalization:** Panel B introduces a critical qualification to this depth hypothesis. When total parameter allocations are held constant across configurations (-$SP$ models), models of vastly different layer depths display highly similar perplexity paths. This reveals that the depth-proportional extrapolation limits observed in Panel A are fundamentally governed by total expressive network capacity and model scale rather than raw layer count alone.

To some extent, this supports our theoretical perspective from a gradient perspective. The optimization signals derived from processing $m$ parallel sequences of length $L/m$ yield information overlaps matching a singular long-sequence training input of length $L$, offering a firm foundation for future development of generalized scaling laws for long-context extrapolation.

### 4.3.4 Token Selection Dynamics in MS Attention

To evaluate the underlying noise-filtering behavior of our **MS Attention** module, we benchmark token retention behavior across varied long-context validation benchmarks using the SnapKV observation window method and a Mistral base model architecture. We compare **MS Attention** against Full KV-cache allocation (All KV) and SnapKV baselines across downstream long-context understanding tasks spanning QA, text summarization, and multi-key retrieval. Performance trends across these dimensions are captured in Table 7.

Table 7: Downstream task performance across diverse multi-domain long-context benchmarks evaluated under alternate cache retention strategies.

| LLM | Method | NarrQA | Qasper | MF-en | HotpotQA | 2WikiMQA | Musique | GovRep | QMSum | M-News | TREC | TriviaQA | SAMSum | PCount | PRe | Lcc | RB-P |
|-----|--------|--------|--------|-------|----------|----------|---------|--------|-------|--------|------|----------|--------|--------|-----|-----|------|
| Mistral | All KV | 26.82 | 33.06 | 49.28 | 42.77 | 27.33 | 19.27 | 32.85 | 24.25 | 27.06 | 71.0 | 86.23 | 42.98 | 2.75 | 86.98 | 55.51 | 52.88 |
| | SnapKV: 2048 | 25.89 | 32.47 | 48.60 | 41.71 | 27.31 | 18.69 | 28.81 | 24.50 | 26.60 | 70.0 | 86.27 | 42.47 | 3.09 | 87.43 | 55.93 | 52.01 |
| | SnapKV: 4096 | 26.41 | 33.46 | 49.81 | 42.32 | 27.93 | 18.76 | 30.74 | 24.19 | 27.08 | 71.0 | 86.25 | 43.01 | 2.73 | 86.18 | 55.62 | 52.65 |
| | **MS Attention**: 4096 | 26.52 | 33.06 | 49.45 | 42.54 | 27.72 | 19.21 | 32.64 | 24.58 | 26.95 | 72.0 | 86.32 | 43.07 | 5.46 | 87.36 | 56.40 | 52.10 |

## 4.4 Baseline Comparisons and Resource Efficiency

### 4.4.1 Extended Baseline Comparisons

To validate the fairness and competitive strength of our empirical results, we extend our baseline evaluations to include Self-Extend (Jin et al., 2024), Position Interpolation (PI) (Chen et al., 2023), YaRN (Peng et al., 2023), CEPE Yen et al. (2024), DAPE (Zheng et al., 2024b), and an absolute baseline variant denoted as NTK-64K. Perplexity benchmarks across context lengths from 2K to 64K tokens are consolidated in Table 8.

Table 8: PG19 language modeling perplexity (PPL) comparison across extended baselines and varied context lengths.

| FT Window | Method | 2K | 4K | 8K | 16K | 32K | 64K |
|-----------|--------|-----|-----|-----|-----|-----|-----|
| 4K | Self-Extend | 6.91 | 6.52 | 6.44 | 6.38 | 6.28 | 7.45 |
| 32K | PI | 7.21 | 6.81 | 6.56 | 6.45 | 6.25 | - |
| 32K | YaRN | 7.01 | 6.69 | 6.47 | 6.31 | 6.23 | - |
| - | CEPE | 7.19 | 6.87 | 6.86 | 6.71 | 6.54 | 6.41 |
| 64K | NTK-64K | 6.92 | 6.50 | 6.42 | 6.29 | 6.23 | 6.21 |
| 16K | **CRG-NTK** (w/o LoRA) | 6.93 | 6.55 | 6.41 | 6.29 | 6.17 | 6.11 |

Our approach, even evaluated without low-rank adaptation constraints (***CRG-NTK** w/o LoRA*), demonstrates structural superiority. Fine-tuned strictly within a 16K window, our method outputs lower perplexity metrics at 32K and 64K lengths than baseline methods requiring explicit 32K or 64K sequence length fine-tuning (such as YaRN, PI, or NTK-64K).

In addition to PPL, we conducted additional comparisons of Self-Extend and DAPE. For Self-Extend, we follow the evaluation protocol of NTK-64K Lu et al. (2024). Due to differences in evaluation methodology (we test by concatenating all sequences without padding), our reproduced results for NTK-64K (denoted NTK-64K*) differ slightly from the original report. Table 9 summarizes the comparison across perplexity (PPL), needle retrieval, LongBench, and RULER.

Table 9: Comparison with training-free and fine-tuned baselines on long-context tasks.

| Attention / Method | PPL | Needle | LongB | RULER |
|---|---|---|---|---|
| *Approximate Attention (Frozen)* | | | | |
|    LM-Infinite | 6.71 | 23.9 | 25.84 | 12.34 |
|    Self-Extend | 6.11 | 25.8 | 33.62 | 29.50 |
| *Exact Attention (Frozen)* | | | | |
|    NTK-F | 14.52 | 18.8 | 25.54 | 0.72 |
| *Fine-Tuned* | | | | |
|    PI | 5.85 | 42.1 | 33.48 | 57.66 |
|    YaRN | 5.85 | 46.7 | 33.45 | 36.95 |
|    CLEX | 5.82 | 71.1 | 33.48 | 52.17 |
|    NTK-32K | 5.79 | 83.7 | 35.32 | 59.42 |
|    NTK-64K | 5.93 | 69.1 | 34.30 | 60.03 |
|    NTK-64K* (our repro.) | 6.12 | 71.2 | 33.79 | 62.46 |
|    **CRG-NTK (Ours)** | **6.19** | **94.5** | **36.51** | **69.27** |

For DAPE, we trained our model on the Book3 dataset with a 512 token length following the original DAPE setup. Although we could not use exactly the same data and hyperparameters, we report the extrapolation performance in terms of perplexity. Table 10 shows that our CRG-NTK+MS achieves significantly lower perplexity and better extrapolation than DAPE, demonstrating the advantage of dynamic growth and selective attention.

Table 10: Comparison with DAPE on Book3 extrapolation (PPL).

| Method | 512 | 1024 | 2048 | 4096 | 8192 |
|---|---|---|---|---|---|
| DAPE | 19.22 | 18.22 | 17.15 | 17.63 | 17.88 |
| CRG-NTK + MS | 12.53 | 11.76 | 10.92 | 10.56 | 10.98 |

### 4.4.2   Training and Inference Efficiency

Table 11 compares the fine-tuning time of our method with LongLoRA and LongRoPE. Our approach significantly reduces both the memory overhead and fine-tuning time. Due to the strong extrapolation capability of our method, it requires only short-length fine-tuning to achieve the same context window length as existing methods. As shown in the table, we fine-tune with a sequence length of 64K for only 32 hours, yet successfully generalize to a 1M-length context window. For LLaMA2-7B with 32 layers, the theoretical extrapolation limit implies that a training length of 32K is sufficient. However, since performance degradation may occur near the extrapolation boundary, we include a small margin by slightly increasing the training length. For LongRoPE its expansion to 1M length requires 8 A100s for 5 days of search (data from its paperDing et al. (2024)) and needs to be fine-tuned on 128K length, so its total time is $5 \times 24h \times 8 + 101h$, where 101h comes from the fine-tuning time required to perform 128K for our configurations, and the LongRoPE method, since it does not use LoRA and MS Attention would only be longer. Additionally, when

Table 11: Training hours comparison on LLaMA2-7B for various positional encoding methods to realize a certain context window length.

| Model | Up to 64K | Up to 256K | Up to 512K | Up to 1024K | Up to 2048K |
|---|---|---|---|---|---|
| LongLora | 52.4h×8 | OOM | OOM | OOM | OOM |
| LongRoPE | – | 3×24h×1 | 3×24h×2 | 5×24h×8 + 98h | 5×24h×8 + 202h |
| CRG NTK | 2.6h×1 | 5.9h×1 | 13.8h×1 | 29.2h×1 | 64.7h×1 |

performing inference with MS Attention, its inference speed remains nearly constant and is approximately half that of a sliding window with a fixed window size.

To establish a transparent assessment of operational overhead, we log the computational footprints required to fine-tuning with 128K. We report performance metrics across two core hardware descriptors: total cumulative GPU-Hours (normalized to NVIDIA A100 environments) and raw Wall-Clock deployment time. Fixed parameter optimization bounds are benchmarked against LongLoRA (Chen et al., 2024), YaRN (Peng et al., 2023), and LongRoPE (Ding et al., 2024) in Table 12.

Table 12: Hardware overhead and resource efficiency tracking for 128K context generalization limits on NVIDIA A100 infrastructure.

| Method Framework | Fine-tuning Length | GPU-Hours (A100) |
|---|---|---|
| LongLoRA | 128K | 186 |
| YaRN | 128K | 372 |
| LongRoPE (Search Included) | 128K | 1162 |
| LongRoPE (Fine-tune Only) | 128K | 202 |
| **CRG-NTK (Ours)** | 64K | **63** |
| **CRG-NTK (Ours)** | 128K | **149** |

Our approach yields substantial resource savings compared to alternative pipelines. **CRG-NTK** completely avoids the costly evolutionary search phases required by LongRoPE, which demand an additional 960 GPU-hours before fine-tuning can begin. When optimizing directly at the 128K boundary, our method consumes only 149 GPU-hours, demonstrating superior efficiency over both LongRoPE (202) and YaRN (372). This efficiency stems from the synergy between our low-rank parameter paths and the memory preservation characteristics of the **MS Attention** module. In terms of wall-clock time, our 64K context fine-tuning pipeline requires approximately 32 hours on a compact node arrangement of $2 \times$ A100 GPUs, while the full 128K expansion finishes within 101 hours.

### 4.4.3 Inference Cost of MS Attention

We measured the inference latency of MS Attention (selecting 8K KV entries) against FlashAttention (Full Attention) on a 16-layer model. As shown in Table 13, MS Attention has similar latency to Full Attention for lengths up to 8K. Between 16K and 32K, the selection and merge steps introduce a small overhead. Beyond 64K, MS Attention becomes increasingly faster, achieving a 3.5× speedup at 262K length. This table complements the efficiency story presented in the main text.

### 4.5 Long-Context Retrieval and Standard Benchmarks

### 4.5.1 Long Context Retrieval Performance

In this subsection, we evaluated the performance of our context extension on the Ruler test set. We fine-tuned parameters using the same CRG NTK configuration, with fine-tuning lengths of 16K and 64K, respectively. The test length was set to 16K and extrapolated to 100K, as shown in Table 14.

To evaluate limits under true long-context scaling conditions, we also implement our structural framework within the Llama3-8B architecture. Following a standard setup comparable to LongRoPE2, the model is

Table 13: Inference latency (ms per token) comparison: MS Attention (select=8K) vs. FlashAttention (Full Attention).

| Sequence Length | FlashAttention (ms) | MS Attention (ms) |
|---|---|---|
| 8,192 | 47.69 | 50.66 |
| 16,384 | 117.51 | 126.27 |
| 32,768 | 331.92 | 355.71 |
| 65,536 | 1057.79 | 782.67 |
| 131,072 | 3801.54 | 1759.15 |
| 262,144 | 14420.34 | 4087.24 |

Table 14: Long Context Retrieval Performance (Ruler)

| Task | llama2-7B-chat (16K) | Ours (FT: 16K) | Ours (Test length 100K) |
|---|---|---|---|
| niah_multiquery | 0.1444 | 0.9855 | 0.8350 |
| niah_multivalue | 0.1267 | 0.9675 | 0.8075 |
| niah_single_1 | 0.1475 | **1.0000** | **1.0000** |
| niah_single_2 | 0.1225 | **1.0000** | **1.0000** |
| niah_single_3 | 0.1250 | **1.0000** | 0.9983 |
| ruler_cwe | 0.2500 | 0.7404 | 0.4710 |
| ruler_fwe | 0.3905 | 0.7067 | 0.4900 |
| ruler_qa_hotpot | 0.0271 | 0.3860 | 0.3100 |
| ruler_qa_squad | 0.0331 | 0.5645 | 0.4850 |

fine-tuned at a 128K sequence length utilizing a 10B token resource budget. Retrieval accuracy profiles are measured on RULER at 128K (interpolation) and 256K (extrapolation) tokens, and benchmarked directly against contemporary configurations in Table 15.

Table 15: RULER benchmark accuracy comparisons on Llama3-8B across interpolation (128K) and extrapolation (256K) text limits.

| Method | RULER 128K | RULER 256K |
|---|---|---|
| YaRN | 49.39 | — |
| NTK Baseline | 73.19 | 72.92 |
| LongRoPE | 73.40 | — |
| LongRoPE2 | 82.03 | 74.79 |
| **CRG-NTK (Ours)** | **83.91** | **80.36** |

Under perfectly normalized training boundaries, **CRG-NTK** outperforms fixed-ratio scaling schemas, establishing a retrieval accuracy score of 83.91% at the 128K fine-tuning limit. Crucially, when evaluating true extrapolation capabilities at 256K tokens, our dynamic scaling mechanism retains an accuracy profile of 80.36%, outperforming LongRoPE2 by a significant 5.57 percentage points. This confirms the advantages of dynamic-ratio scaling over static interpolation pipelines.

### 4.5.2 Short-Context Knowledge Preservation

A common phenomenon in context extension is the degradation of base model performance on short-sequence benchmarks such as MMLU, as shown in 16. This is because changes in positional encoding affect the original modeling, and fine-tuning on extremely long sequences dilutes the fitting distribution for short-sequence modeling. Consequently, many methods require fine-tuning on specific short-sequence datasets to restore short-sequence modeling capabilities when expanding the context window. In contrast, our CRG NTK achieves performance comparable to theirs without fine-tuning on short-sequence datasets.

Furthermore, we believe this drop is also driven by the lower data quality of the long-form text corpora utilized during fine-tuning (e.g., SlimPajama). To address this, we introduce a secondary high-quality Supervised Fine-Tuning (SFT) phase spanning 1B high-quality tokens, matching standard alignment protocols in the literature. Evaluation benchmarks across baseline and aligned configurations for Llama2-7B are provided in Table 17.

Table 16: Performance comparison of different 7B parameter models on short standard benchmarks.

| Model | ARC-c | HellaSwag | MMLU | TruthfulQA |
|---|---|---|---|---|
| Original LLaMA2-7B | 53.1 | 78.6 | 46.6 | 39.0 |
| Together | 47.6 | 76.1 | 43.3 | 39.2 |
| Code LLaMA | 42.4 | 64.8 | 40.1 | 37.1 |
| YaRN ($s = 16$) | 52.4 | 78.7 | 42.4 | 38.2 |
| YaRN ($s = 32$) | 52.2 | 78.5 | 41.8 | 37.4 |
| LongRoPE-2048k (ft=128k) | 52.9 | 76.5 | 43.4 | 38.8 |
| LongRoPE-2048k (ft=256k) | 51.0 | 75.3 | 39.6 | 37.3 |
| **LLaMA2-7B-MS (ft=16K, CRGNTK)** | 51.9 | 77.8 | 41.9 | **40.1** |

Table 17: Short-context knowledge verification on MMLU across base, extended, and high-quality alignment (SFT) variants using Llama2-7B.

| Model Framework | Fine-tuning Corpus | Tokens | MMLU Score |
|---|---|---|---|
| Llama2-7B (Original Base) | — | 0B | **46.6** |
| Llama2-7B (Continued FT) | SlimPajama | 2B | 42.7 |
| Llama2-7B (Short-SFT Base) | High-Quality Corpus | 1B | 44.2 |
| *Extended Variants (No SFT)* | | | |
|    YaRN | SlimPajama | 2B | 41.5 |
|    LongRoPE | SlimPajama | 2B | 42.0 |
|    **CRG-NTK** | SlimPajama | 2B | 41.9 |
| *Aligned Variants (With SFT)* | | | |
|    YaRN + SFT | High-Quality Corpus | 1B | 43.9 |
|    LongRoPE + SFT | High-Quality Corpus | 1B | 43.4 |
|    **CRG-NTK + SFT** | High-Quality Corpus | 1B | **43.7** |

The empirical results show that the initial drop in MMLU accuracy is an inherent characteristic of long-text fine-tuning data rather than a flaw in the positional scaling mechanics, as continuing base model training on raw SlimPajama tokens drives an identical reduction (42.7). Following the integration of our secondary targeted SFT phase (**CRG-NTK + SFT**), short-context performance is effectively restored to 43.7, matching or exceeding competitive baselines.

## 4.6 Pretraining, LongBench, and Extrapolation Analysis

Table 27 shows Few-shot Learning and Code Completion Evaluation on LongBench: These tasks do not require chat instruct fine-tuning. We have observed significant improvements in "trec", "triviaqa", "lcc", and "repobench-p" using our method on Llama2-7B-4k comparing with Llama2-7B-chat-4k, especially outperforming other models in the LCC task.

Tables 19 and 20 show the results of pretraining with MS Attention on PG19 and ImageNet, where our method outperforms various baselines. For PG19, test data is truncated when it is larger than the length of the test.

We further conducted experiments on automatic length extrapolation. After training on sequences of 8K and 16K tokens, our method shows stable perplexity even when extending inference to 8x the training length. Table 21 presents the results.

Table 18: Few-shot Learning and Code Completion Evaluation on LongBench

| Model | TREC | TriviaQA | SAMSum | LCC | RepoBench-P |
|---|---|---|---|---|---|
| GPT-3.5-Turbo-16k | 68 | **91.4** | **41.7** | 54.7 | 53.6 |
| LongChat-v1.5-7B-32k | 63.5 | 82.3 | 34.2 | 53 | **55.3** |
| XGen-7B-8k | 65.5 | 77.8 | 25.3 | 38.6 | 38.6 |
| InternLM-7B-8k | 52 | 77.8 | 21.2 | 44.1 | 28.8 |
| ChatGLM2-6B-32k | 62.5 | 78.7 | 36.3 | 55.6 | 49.9 |
| Vicuna-v1.5-7B-16k | 71.5 | 86.2 | 40.8 | 51 | 43.5 |
| ChatGLM3-6B-32k | **79** | 87.1 | 38.2 | 57.66 | 54.76 |
| **Llama2-7B-chat-4k** | 61.5 | 77.8 | 40.7 | 52.4 | 43.8 |
| **Llama-7B-32k-MS-PI** | 72.6 | 85.7 | 40.6 | 61.95 | 49.09 |
| **Llama-7B-16k-MS-CRGNTK** | 72.6 | 87.5 | 39.3 | **65.24** | 54.76 |
| **Mistral-7B-16k-MS-CRGNTK** | 65.6 | 88.3 | 38.3 | 64.4 | 54.1 |

Table 19: Perplexity (PPL) results on the PG19.

| Model | Params | PPL (PG19) |
|---|---|---|
| TransformerXL | - | 36.3 |
| Routing Transformer | - | 33.3 |
| Landmark Attention-200M | 200M | 14.55 |
| Selection-Merging Attention-200M | 200M | 10.89 |

Table 20: Top-1 accuracy results on ImageNet.

| Model | FLOPs (G) | Top-1 Accuracy (%) |
|---|---|---|
| VVT-T-12.9M | 2.0 | 79.4 |
| Swin Transformer-29M | 4.2 | 81.3 |
| Biformer-13.1M | 2.2 | 81.4 |
| Selection-Merging Attention-13.1M | 2.1 | 82.1 |

Table 21: Perplexity across different sequence lengths for Select-Merge Attention. Test data is truncated when it is larger than the length of the test.

| Method | 8K | 16K | 32K | 64K | 128K |
|---|---|---|---|---|---|
| Select-Merge Attention (8K) | 10.9 | 10.57 | 12.57 | 14.16 | - |
| Select-Merge Attention (16K) | 9.82 | 9.56 | 10.93 | 12.16 | 16.28 |

### 4.7 Failure Modes and Trade-offs

We provide an explicit calibration of where CRG-NTK excels and where it faces limitations. The choice between MS Attention and Full Attention during inference creates a task-dependent trade-off:

**MS Attention** excels at *reasoning-heavy* long-context tasks and extreme-length extrapolation, as its selective KV filtering suppresses noise in ultra-long sequences. However, its retrieval precision is limited without large-scale training (e.g., 10B+ tokens), because the sparse selection mechanism may discard task-critical KV pairs. This manifests as underperformance on needle-in-a-haystack and Ruler-style retrieval when fine-tuned on limited data (Table 5).

**Full Attention** is optimal for *retrieval-heavy* tasks, achieving superior needle retrieval and Ruler scores (69.1 vs. 47.6 at 128K). Its limitation is weaker extrapolation: at extreme lengths, the full sequence's noise floor degrades generation quality, causing PPL to rise more sharply than MS Attention (7.42 vs. 6.86 at 1M).

**Practical recommendation.** Use MS Attention during fine-tuning for efficiency and for inference on reasoning tasks at extreme lengths; switch to Full Attention for retrieval-critical applications or when fine-tuning budget permits full attention adaptation.

## 5 Conclusion

In this paper, we build upon the LongLoRA framework, utilizing a mechanism of selection and merging within the Attention mechanism (referred to as MS Attention). By employing our approach on $1 \times$ A100 80GB GPUs, we achieve the same level of length extension for Llama2-7B as LongLoRA and LongRoPE

does on $8 \times$ A100 80GB GPUs. Specifically, we extend the context length of Llama2-7B to 2M tokens, significantly reducing the resource requirements for handling long sequences. Moreover, We also indicate the possible extrapolation limit of the model: the number of layers of the model. Finally, the flexibility of our MS Attention mechanism allows for adjustable selection size and selection quantity, which, combined with restricting the attention range of each token, enables adaptable fitting and convergence rates.

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

# A   Additional Ablation Experiments

Additionally, we highlight the advantages of our efficient Attention in pre-training. Our MS Attention method is capable of breaking the translation invariance of Full Attention and intrinsically achieving high-multiple length extrapolation. For example, we were able to achieve excellent performance using MS Attention and a context window length of 8K for training on PG19, and then tested on a length of 64K without a perplexity explosion, as shown in Tables 19 and Tables 21. In addition, we were able to achieve performance comparable to the current optimal efficient Attention by pre-training on Imagenet, as shown in Table 20. Finally we include the results of the tests on LongBench, as shown in Table 27.

Table 22: PPL on the PG19 with Lora or not.

| Model Configuration | 2K | 4K | 8K | 16K | 32K |
|---|---|---|---|---|---|
| Llama-7B-MS-no_LoRA | 7.06 | 6.96 | 6.78 | 6.56 | 6.93 |
| Llama-7B-MS-LoRA | 7.61 | 7.42 | 7.17 | 7.03 | 7.06 |

Table 23: Memory and Time w/wo DeepSpeed and LoRA

| Configuration | Memory (MB) | Time (s/it) |
|---|---|---|
| With DeepSpeed & LoRA | 36,786 | 6.37 |
| Without DeepSpeed | 31,524 | 6.19 |
| Without LoRA | 35,074 | 7.28 |
| Without DeepSpeed & LoRA | 40,132 | 6.38 |

Table 24: PPL on the PG19 (50 samples) with MS Attention or not. FT steps: 1000, FT length: 16384, ntk scale: 16384.

| Model Configuration | 16K | 64K | 128K | 256K (scale=16384/32768/8192) |
|---|---|---|---|---|
| Llama-7B-MS-ntk16384 | 9.58 | 9.55 | 22.09 | 55.76/90.92/181.20 |
| Llama-7B-FA-ntk16384 | 9.57 | 9.36 | 21.54 | 53.92/103.03/298.86 |
| Llama-7B-MS-PI16 | 9.23 | 9.18 | - | - |
| Llama-7B-FA-PI16 | 9.18 | 9.04 | - | - |

Table 25: Ablation experiment for PEFT. PEFT steps: 1000, Evaluation DataSet: PG19 validation.

| PEFT | Training Context Length w/wo setting | Evaluation Context Length | | |
|---|---|---|---|---|
| | | 32768 | 16384 | 8192 |
| qkvo | 16384-m16-s64 | 8.96 | 8.33 | 8.32 |
| ko | 16384-m16-s64 | 8.55 | 8.36 | 8.43 |

Table 26: PPL for Llama2-7B with MS Attention on PG19.

| Parameters | PPL |
|---|---|
| **MS-patch16-select512-merge64** | 7.17 |
| **MS-patch16-select64-merge4** | 6.95 |
| **MS-patch16-select16-merge4** | 6.83 |
| **MS-patch8-select32-merge8** | 6.18 |
| **MS-patch8-select32-merge4** | 6.09 |

# B   Related Work

**Efficient Attention Mechanisms** To fully exploit the inherent sparsity and positional relationships between tokens, a significant body of research has focused on developing efficient attention mechanisms. These mechanisms reduce the computational complexity of attention operations by focusing on a subset of tokens at each step, thus processing long sequences more efficiently or reducing resource consumption. Existing methods first preserve local features and then use various strategies to attend to more distant tokens. For instance, BigBirdZaheer et al. (2020) and PerformerChoromanski et al. (2022) use random patterns, LongformerBeltagy et al. (2020) and DETRZhu et al. (2021) use fixed patterns, while BiformerZhu et al. (2023) and Routing TransformerRoy et al. (2021) utilize relevance routing mechanisms. Our proposed relevance selection and merging mechanism adapts flexibly to various scenarios and is compatible with FlashAttention2Dao et al. (2022), achieving more efficient and general sparse attention.

**Positional Encoding** Another research direction aimed at extending sequence length in LLMs focuses on positional encoding techniques, such as positional interpolation and extrapolation. Most pretrained models are trained on fixed-length sequences with fixed positional encodings, leading to performance degradation when extended to unknown positions. Therefore, numerous studies have analyzed the impact of positional

Table 27: Few-shot Learning and Code Completion Evaluation on LongBench

| Model | TREC | TriviaQA | SAMSum | LCC | RepoBench-P |
|---|---|---|---|---|---|
| GPT-3.5-Turbo-16k | 68 | **91.4** | **41.7** | 54.7 | 53.6 |
| LongChat-v1.5-7B-32k | 63.5 | 82.3 | 34.2 | 53 | **55.3** |
| XGen-7B-8k | 65.5 | 77.8 | 25.3 | 38.6 | 38.6 |
| InternLM-7B-8k | 52 | 77.8 | 21.2 | 44.1 | 28.8 |
| ChatGLM2-6B-32k | 62.5 | 78.7 | 36.3 | 55.6 | 49.9 |
| Vicuna-v1.5-7B-16k | 71.5 | 86.2 | 40.8 | 51 | 43.5 |
| ChatGLM3-6B-32k | **79** | 87.1 | 38.2 | 57.66 | 54.76 |
| **Llama2-7B-chat-4k** | 61.5 | 77.8 | 40.7 | 52.4 | 43.8 |
| **Llama-7B-32k-MS-PI** | 72.6 | 85.7 | 40.6 | 61.95 | 49.09 |
| **Llama-7B-16k-MS-CRGNTK** | 72.6 | 87.5 | 39.3 | **65.24** | 54.76 |
| **Mistral-7B-16k-MS-CRGNTK** | 65.6 | 88.3 | 38.3 | 64.4 | 54.1 |

Table 28: Perplexity results on proofpile dataset. Fine-tuning with MS Attention; Testing with MS Attention or Full Attention.

| Model | 4K | 8K | 32K | 64K | 128K | 256K |
|---|---|---|---|---|---|---|
| Llama2-7B-LongLora (ft=32K, PI) | 3.01 | 2.78 | 2.58 | 2.57 | - | - |
| Llama2-7B-LongRoPE (ft=256K) | 3.71 | 3.50 | 2.60 | 2.36 | 2.26 | 1.88 |
| Llama2-7B-MS (ft=16K, CRG NTK, test:FA) | 3.12 | 2.76 | 2.56 | 2.58 | 2.60 | 2.68 |
| Llama2-7B-MS (ft=16K, CRG NTK, test:MS) | 2.96 | 2.64 | 2.32 | 2.31 | 2.26 | 2.28 |

Table 29: Comprehensive evaluation of our method compared to baselines on LLaMA2-7B and LLaMA2-13B models. PPL: Perplexity (lower is better), FA: Full Attention, MS: MS Attention.

| Model | Method | FT Length | PPL 8K | PPL 32K | PPL 64K | PPL 128K |
|---|---|---|---|---|---|---|
| | Codellama NTK | 16K | 3.71 | 2.74 | 2.55 | 2.71 |
| LLaMA2-7B | YaRN | 64K | 3.51 | 2.65 | 2.42 | > 10 |
| | **Ours (FA/MS)** | **16K** | **3.12** | **2.56** | **2.58 / 2.32** | **2.60 / 2.26** |
| | Codellama NTK | - | 3.54 | 2.63 | 2.41 | 2.54 |
| LLaMA2-13B | YaRN | - | 3.25 | 2.50 | 2.29 | > 10 |
| | YaRN | - | 3.29 | 2.53 | 2.31 | 2.24 |
| | **Ours (FA/MS)** | - | **3.05** | **2.42** | **2.30 / 2.22** | **2.31 / 2.16** |

| Method (13B) | ARC-c | HellaSwag | MMLU | TruthfulQA |
|---|---|---|---|---|
| LLaMA 2 | 59.4 | 82.1 | 55.8 | 37.4 |
| Code LLaMA | 40.9 | 63.4 | 32.8 | 43.8 |
| YaRN (s = 16) | 58.1 | 82.3 | 52.8 | 37.8 |
| YaRN (s = 32) | 58.0 | 82.2 | 51.9 | 37.3 |
| **CRG NTK (Ours)** | **58.5** | 81.2 | 52.1 | **40.7** |

encodings and modified them through interpolation or extrapolation to extend to longer sequences. For example, Position InterpolationChen et al. (2023), NTK-awareNtk (2023), YarnPeng et al. (2023), and LongRopEDing et al. (2024) mitigate the effects of pretrained positional encodings by using interpolation with different scales based on frequency importance, effectively extending sequence modeling lengths.

**Efficient Fine-Tuning** Efficient fine-tuning of LLMs has become a critical research direction, especially for handling long sequences. Techniques like Input-tuningAn et al. (2022) and LoRAHu et al. (2021) have shown significant promise in this area. Building on the LongLoraChen et al. (2024) method, we further optimize by using fewer parameters for fine-tuning, aiming to reduce the computational and memory overheads associated with fine-tuning large models on extended sequences while maintaining their performance on downstream tasks.

# C  Positional Awareness and Breaking Translation Invariance

## C.1  Positional Awareness:

Our algorithm( 3.2, 4, E.2) leverages semantic token-based routing and selection mechanisms to capture boundary effects, thus disrupting the translation invariance inherent in global Attention mechanisms.

Semantic Token Routing and Selection: The routing process in our algorithm employs semantic tokens, which represent the semantics of a region, carrying contextual information similar to a sliding window in convolutional neural networks (CNNs). Due to the contextual variance surrounding each token, the derived semantic tokens differ, leading to the routing of different KV tokens. This results in the selection of distinct interpolation variables and, consequently, varying outputs. This mechanism allows our approach to effectively capture boundary effects, unlike traditional global Attention which maintains translation invariance.

### C.2 Extrapolation through Finetuning and Position Interpolation:

Our method can be fine-tuned during training and later applied to full Attention during inference, enabling significantly higher extrapolation factors. Specifically, our MS Attention mechanism can achieve up to twice the extrapolation compared to traditional methods, with potential reasons outlined as follows:

- **Improved Relative Positional Awareness:** The primary reason behind this extrapolation capability is our selection mechanism, which better extends the model's awareness of relative positions. In full Attention, tokens near the boundaries rely on all preceding tokens for regression, potentially leading the model to integrate all previous positional information. In contrast, our selection mechanism integrates a subset of positions into the final positional information, leading to a more accurate representation of relative positions. This method allows the model to assert the correct positional information even with different combinations of preceding positions.

- **Extrapolation Limits:** The reason for successful extrapolation up to twice the training length, but not beyond, lies in the finetuning process itself. During finetuning with a length $l$, the model learns the correct positional information and its integration within this range. Therefore, the model can correctly extrapolate up to $2l$, but further extrapolation may fail as it exceeds the recognized range.

For instance, when finetuning LLaMA2-7B with a length of $l = 16K$, the positional range for 'Position_ids' is set from 1 to 16K. By increasing the initial position range, e.g., choosing starting positions within the 1-64K range ($l_1 = 64K$), and setting the interpolation ratio for position interpolation as $80K/4K = 20$ or higher, we successfully achieve 100% accuracy on the 80K-length passkey task. Similarly, by exposing the model to even longer positions and using our adaptive positional interpolation encoding, we set $l_1 = 128K$ and successfully extrapolate to 144K. This analysis suggests that our method, even with simple position interpolation, has the potential to achieve near-infinite length extrapolation.

### C.3 Extrapolation through Finetuning and NTK-Based Positional Encoding:

Combining our MS Attention with NTK-based positional encoding enables extrapolation by factors exceeding tenfold. This approach is explained using the radix theory proposed by the authors of RoPE:

- **Learning Relative Magnitudes:** Our MS Attention accurately learns the relative magnitudes of positional encodings. By incrementally increasing the radix base size, the model learns to represent the relative magnitudes across different radices. Additionally, by varying the starting positions, the model further refines its understanding of relative magnitudes within the same radix.

- **Future Work:** Future extensions will incorporate relative shifts between Q tokens and K tokens, allowing the model to sense deviations between different positions of Q and K.

This combination of MS Attention and NTK-based encoding showcases a significant potential for enhancing extrapolation capabilities, ultimately pushing the boundaries of positional understanding in large language models.

## D  Position Information Learning Process Analysis:

### D.1  Forward Pass

In the computation process, the steps of Attention typically follow this sequence:

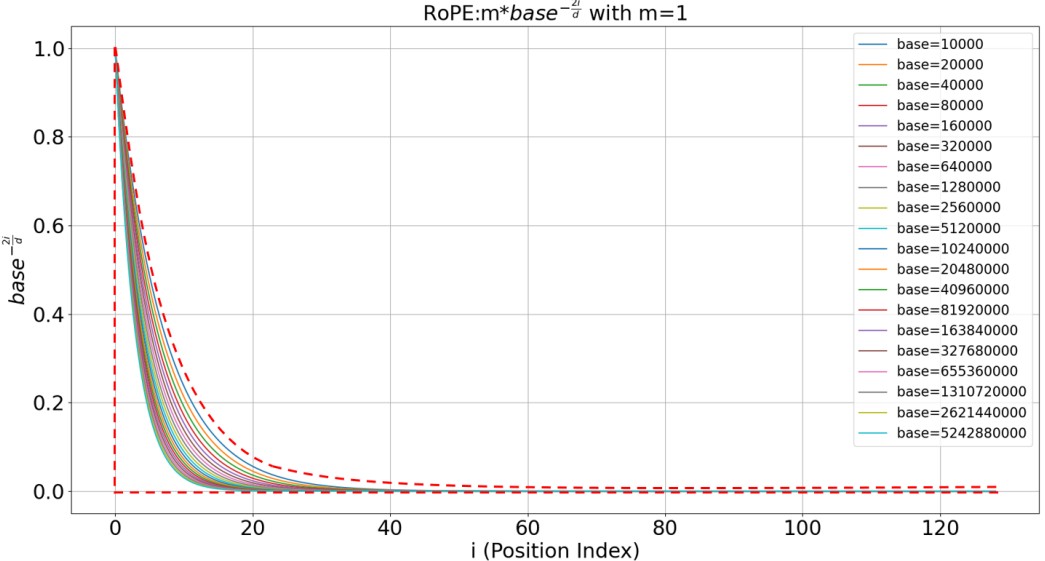

Figure 2: **Dynamic scanning PE solution space.**

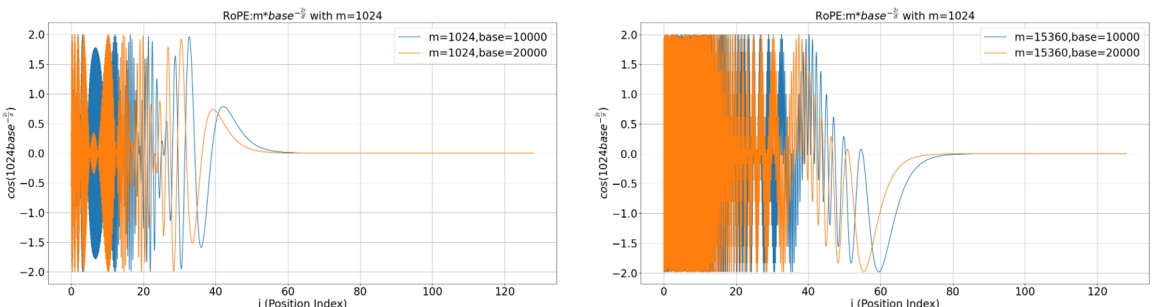

Figure 3: **Differences in information encoded in different scale positions.**

1. First, we introduce two components, $P_c$ and $P_s$, to model cosine and sine positional encoding:

$$K_P = K \odot P_c + K_s \odot P_s$$

$$Q_P = Q \odot P_c + Q_s \odot P_s$$

The matrices $P_c$ and $P_s$ are structured as:

$$P_c = \begin{bmatrix} \cos \theta_0 & \cos \theta_0 & \dots & \cos \theta_{\frac{d}{2}-1} \\ \cos 2\theta_0 & \cos 2\theta_0 & \dots & \cos 2\theta_{\frac{d}{2}-1} \\ \vdots & & \vdots & \\ \cos n\theta_0 & \cos n\theta_0 & \dots & \cos n\theta_{\frac{d}{2}-1} \end{bmatrix}$$

$$P_s = \begin{bmatrix} \sin \theta_0 & \sin \theta_0 & \dots & \sin \theta_{\frac{d}{2}-1} \\ \sin 2\theta_0 & \sin 2\theta_0 & \dots & \sin 2\theta_{\frac{d}{2}-1} \\ \vdots & & \vdots & \\ \sin n\theta_0 & \sin n\theta_0 & \dots & \sin n\theta_{\frac{d}{2}-1} \end{bmatrix}$$

Moreover, we apply a learned matrix $W_s$ to $K_s$ as follows:

$$K_s = K \times W_s = K \times \begin{bmatrix} 0 & -1 & & & 0 \\ 1 & 0 & & & \\ & & \ddots & & \\ & & & 0 & -1 \\ 0 & & & 1 & 0 \end{bmatrix}_{n \times n}$$

2. Compute similarity scores:

$$\text{Scores} = \frac{Q_P K_P^\top}{\sqrt{d_k}}$$

3. Apply Softmax:

$$\text{Weights} = \text{softmax}(\text{Scores})$$

4. Compute the output:

$$\text{Output} = \text{Weights} \times V$$

## D.2 Theoretical Analysis of Positional Encoding Extrapolation Based on Linear Approximation

### D.2.1 Notation and setup

Let $d$ denote the model hidden dimension (assumed even) and $m := d/2$ the number of 2D rotational blocks used by RoPE-style position encoding. Let base $> 1$ denote the RoPE base. For block index $i \in \{0, \ldots, m-1\}$ we define the block frequency parameter

$$\theta_i = \text{base}^{-2i/d},$$

and the phase at position $p$ as

$$\phi_i(p) = p\theta_i.$$

We consider a Transformer trained on sequences of length $M$ (positions $0, \ldots, M-1$) with parameters $\Theta$ minimizing

$$\min_{\Theta} \mathcal{L}(\Theta; \mathcal{D}_M).$$

We are interested in extrapolation to length $L > M$, and in finding (possibly small) parameter adjustments $\Delta\Theta$ or deterministic transforms on positions/frequencies that preserve the loss behavior:

$$\mathcal{L}(\Theta + \Delta\Theta; \mathcal{D}_L) \approx \mathcal{L}(\Theta; \mathcal{D}_M).$$

### D.2.2 Linear approximation of attention and positional terms

We adopt a first-order / linearized viewpoint of the attention exponential term. For queries $\mathbf{q}_n$ and keys $\mathbf{k}_m$ we approximate:

$$e^{\mathbf{q}_n \mathbf{k}_m^\top} \approx e^{\mathbf{q}_n \mathbf{k}_m^\top}\left(1 + f_p(\mathbf{q}_n, \mathbf{p}(n)) f_p(\mathbf{k}_m, \mathbf{p}(m))^\top - \mathbf{q}_n \mathbf{k}_m^\top\right),$$

where $f_p(\cdot, \cdot)$ denotes the position-dependent linear contribution. For RoPE,

$$f_p(\mathbf{q}_n, \mathbf{p}(n)) f_p(\mathbf{k}_m, \mathbf{p}(m))^\top = \left(Q_i P_c(n) + Q_{i,s} P_s(n)\right)\left(K_j P_c(m) + K_{j,s} P_s(m)\right)^\top,$$

with $P_c(n), P_s(n)$ the cos/sin matrices evaluated at position $n$.

*Proof.* The original attention computation can be expressed as:

$$e^{\frac{1}{\sqrt{d}} \sum_{j=0}^d q_{nj} \cdot k_{sj}} \sum_s e^{\frac{1}{\sqrt{d}} \sum_{j=0}^d q_{nj} \cdot k_{sj} \cdot (\cos[(n-s)\theta_j]-1) + \sum_{j=0}^d \frac{(-1)^{j+1}}{\sqrt{d}} q_{nj} \cdot k_{s,(j+\frac{d}{2})\%d} \cdot (\sin[(n-s)\theta_j])} \tag{17}$$

Let us define the residual term:

$$r(\theta_1, \ldots, \theta_{\frac{d}{2}-1}) = \frac{1}{\sqrt{d}} \sum_{j=0}^{d} q_{nj} \cdot k_{sj} \cdot (\cos[(n-s)\theta_j] - 1) + \sum_{j=0}^{d} \frac{(-1)^{j+1}}{\sqrt{d}} q_{nj} \cdot k_{s,(j+\frac{d}{2})\%d} \cdot (\sin[(n-s)\theta_j]) \quad (18)$$

Since $\theta_j = \text{base}^{-\frac{2j}{d}}$, when base is large, we have the approximations:

$$\cos\theta_j = 1 - \frac{\theta_j^2}{2} + O(\theta_j^4) \quad (19)$$

$$\sin\theta_j = \theta_j + O(\theta_j^3) \quad (20)$$

Following the CRG-NTK method which has been proven to extend context window length and achieve excellent performance, we extend base to very large values (e.g., base $= 2^{40} \times 10000$). In this regime, $\cos\theta_j$ and $\sin\theta_j$ become very small, particularly for dimensions with larger $j$ values. Since $\frac{1}{\sqrt{d}} \sum_{j=0}^{d} q_{nj} \cdot k_{sj}$ is typically on the order of 1, we can apply Taylor expansion to $e^{r(\theta_1, \ldots, \theta_{\frac{d}{2}-1})}$:

$$e^{r(\theta_1, \ldots, \theta_{\frac{d}{2}-1})} = 1 + r(\theta_1, \ldots, \theta_{\frac{d}{2}-1}) + O(r(\theta_1, \ldots, \theta_{\frac{d}{2}-1})^2) \quad (21)$$

$\square$

### D.2.3 Gradient linear-dominance

**Theorem D.1** (Linear dominance of positional gradients). *Under the linearized attention model above and standard smoothness assumptions on activations, the gradient of the loss with respect to positional encoding parameters $P$ (or frequency parameters $\theta_i$) is dominated by the linear positional term:*

$$\frac{\partial \mathcal{L}}{\partial P} \approx \sum_{m,n} \frac{\partial \mathcal{L}}{\partial A_{mn}} \frac{\partial}{\partial P} \big[ f_p(\mathbf{q}_n, \mathbf{p}(n)) \, f_p(\mathbf{k}_m, \mathbf{p}(m))^\top \big],$$

*where $A_{mn}$ denotes pre-softmax attention scores. Consequently, positional modifications that preserve these linear terms approximately preserve gradient directions/magnitudes.*

*Sketch.* The proof follows by substituting the first-order expansion into the attention expression, differentiating the scalar loss $\mathcal{L}(A)$ via chain rule, and observing that higher-order derivatives involving the exponential factor or the quadratic term $\mathbf{q}_n \mathbf{k}_m^\top$ are of smaller magnitude under mild norm bounds on $\mathbf{q}, \mathbf{k}$. Hence the linear position-dependent contribution dominates the derivative w.r.t. positional parameters. $\square$

### D.2.4 Gradient continuity and frequency sensitivity

**Lemma D.1** (Frequency-wise gradient expression). *For RoPE-like encoding, the gradient w.r.t. the $i$-th frequency parameter $\theta_i$ can be written as*

$$\frac{\partial \mathcal{L}}{\partial \theta_i} = \sum_{m,n} \frac{\partial \mathcal{L}}{\partial A_{mn}} \cdot \frac{\partial A_{mn}}{\partial \theta_i},$$

*with (up to the block-specific linear coefficients)*

$$\frac{\partial A_{mn}}{\partial \theta_i} = -(m-n)\Big[ Q_i^{(m)} K_i^{(n)} \sin\big((m-n)\theta_i\big) + Q_{i,s}^{(m)} K_{i,s}^{(n)} \cos\big((m-n)\theta_i\big) \Big].$$

*Sketch.* Differentiate the block-wise rotated representations $R(\phi_i(p))$ with respect to $\theta_i$ where $\phi_i(p) = p\theta_i$. By the product and chain rules, derivatives bring down the multiplicative factor $p$ (hence the difference $m-n$ appears after combining query/key contributions), and the trigonometric derivative yields sine/cosine terms as shown. $\square$

**Remark D.1.** *Lemma D.1 highlights that gradients for different frequency components are weighted by the difference index $(m - n)$ and trigonometric factors; thus, phases beyond the training interval may flip signs or amplify gradients unless controlled.*

### D.2.5 Phase-consistency and gradient-matching conditions

**Definition D.1** (Effective training phase set)**.** *The set of phases exposed during training length $M$ is*

$$\Phi_M \;=\; \{\, m\theta_i \bmod 2\pi : m = 0, \ldots, M - 1, \; i = 0, \ldots, m - 1 \,\}.$$

*The convex hull $\mathrm{conv}(\Phi_M)$ denotes the (real-valued) interval(s) spanned by these phases (we treat phases lifted to the real line under the principal branch approximation).*

**Theorem D.2** (Phase extrapolation condition)**.** *A necessary condition for successful extrapolation (so that gradients and attention statistics behave similarly on length $L$ as on $M$) is that for extrapolated positions $p \in \{M, \ldots, L - 1\}$ the adjusted phases $\phi_i'(p)$ lie in $\mathrm{conv}(\Phi_M)$. Equivalently, under a position-frequency scaling*

$$p\theta_i \mapsto \phi_i'(p) = \frac{p}{s_1} \cdot \frac{\theta_i}{s_2^{2i/d}} = \frac{p}{s_1 s_2^{2i/d}} \theta_i,$$

*we must have for the maximal position $p = L - 1$ that*

$$0 \;\leq\; \frac{L - 1}{s_1 s_2^{2i/d}} \;\leq\; M - 1. \tag{22}$$

*Proof.* If the scaled phases lie outside the training-phase interval, trigonometric factors in Lemma D.1 will take values not seen during training and can cause sign changes or amplitude differences in $\partial A_{mn}/\partial \theta_i$. Hence keeping $\phi_i'(p)$ inside $\mathrm{conv}(\Phi_M)$ (and in particular for $p = L - 1$) ensures the phase-dependent terms remain in the range observed during training. Solving $\phi_i'(L - 1) \leq (M - 1)\theta_i$ yields equation 22. $\qquad\square$

### D.2.6 Variational derivation of scaling factors

We now pose a quantitative objective to choose $(s_1, s_2)$. Define the gradient-discrepancy objective

$$\mathcal{J}(s_1, s_2) \;=\; \sum_{i=0}^{m-1} \left\| \frac{\partial \mathcal{L}}{\partial \theta_i}(L; s_1, s_2) - \frac{\partial \mathcal{L}}{\partial \theta_i}(M) \right\|_2^2.$$

Under the linear approximation (Theorem D.1) the mapping $(s_1, s_2) \mapsto \partial \mathcal{L}/\partial \theta_i(L; s_1, s_2)$ is approximately multiplicative:

$$\frac{\partial \mathcal{L}}{\partial \theta_i}(L; s_1, s_2) \approx \frac{1}{s_1 s_2^{2i/d}} \frac{\partial \mathcal{L}}{\partial \theta_i}(M).$$

Thus minimizing $\mathcal{J}$ w.r.t. $(s_1, s_2)$ reduces approximately to minimizing

$$\widetilde{\mathcal{J}}(s_1, s_2) = \sum_{i=0}^{m-1} \left\| \left( \frac{1}{s_1 s_2^{2i/d}} - 1 \right) \frac{\partial \mathcal{L}}{\partial \theta_i}(M) \right\|_2^2.$$

**Proposition D.1** (Stationary condition for multiplicative approximation)**.** *A stationary (and in some cases minimizer) solution of $\widetilde{\mathcal{J}}$ satisfies approximately*

$$\frac{1}{s_1 s_2^{2i/d}} \approx 1 \quad \text{for those $i$ with large } \left\| \frac{\partial \mathcal{L}}{\partial \theta_i}(M) \right\|_2.$$

*Hence optimal $(s_1, s_2)$ tends to match the most "energetic" frequency components.*

*Sketch.* Take partial derivatives of $\widetilde{\mathcal{J}}$ w.r.t. $s_1, s_2$ and set to zero. The resulting equations emphasize frequencies with large pre-factor norms; when few frequencies dominate, one can approximately match those frequencies, leading to the multiplicative conditions for those indices. $\qquad\square$

### D.2.7 Final algebraic form of the extrapolation constraint

Using $\theta_i = \text{base}^{-2i/d}$, condition (22) can be rewritten as

$$\frac{L-1}{s_1} \leq (M-1)\, s_2^{2i/d} \quad \Longleftrightarrow \quad \frac{(L-1)/s_1}{(\text{base}/s_2)^{2i/d}} \leq \frac{M-1}{\text{base}^{2i/d}}.$$

Therefore the (conservative) target constraint for extrapolation becomes

$$\boxed{\frac{(L-1)/s_1}{(\text{base}/s_2)^{\frac{2i}{d}}} \in \left[0,\; \frac{M-1}{\text{base}^{\frac{2i}{d}}}\right]} \tag{23}$$

for every frequency index $i$ that we require to be "covered" by training. Note that equality corresponds to exact matching of the maximal phase; the interval allows any scaled maximal phase within the training-covered range.

### D.2.8 Analysis of typical extrapolation strategies

**Position interpolation (PI).** Set $s_2 = 1$. Then condition equation 22 reduces to $(L-1)/s_1 \leq M-1$. Choosing

$$s_1 = \frac{L-1}{M-1} \approx \frac{L}{M}$$

maps the target maximum position to the training maximum position and (under multiplicative gradient approximation) yields exact gradient-amplitude matching across all frequencies (Theorem D.1). PI is therefore the uniform, parameter-free approach to preserve gradient amplitudes.

**Frequency/base scaling (NTK-aware-like).** Set $s_1 = 1$. For each frequency $i$, exact matching requires

$$s_2^{2i/d} = \frac{L-1}{M-1} \quad \Longrightarrow \quad s_2 = \left(\frac{L-1}{M-1}\right)^{\frac{d}{2i}}.$$

Since the RHS depends on $i$, a single scalar $s_2$ cannot match all frequencies. In practice, choosing $s_2 > 1$ biases the transform to preserve higher-frequency gradient amplitudes (large $i$), which is sometimes desired when high-frequency structure carries important sequence-local information.

**Hybrid and optimization-based calibration.** Treat $(s_1, s_2)$ as tunable and directly optimize $\mathcal{J}(s_1, s_2)$ on a small calibration set (or via projected gradient descent). This hybrid method can trade off which frequency bands to prioritize according to their gradient energy, and is theoretically justified by the variational derivation above.

### D.3 Algorithm: projected gradient calibration on $(s_1, s_2)$

---
**Algorithm 1** Calibrate $(s_1, s_2)$ via projected gradient descent

---
1: initialize $s_1 \leftarrow 1$, $s_2 \leftarrow 1$
2: **for** $t = 1, 2, \ldots$ **do**
3:     compute $\Delta_i = \partial \mathcal{L}/\partial \theta_i (L; s_1, s_2) - \partial \mathcal{L}/\partial \theta_i (M)$ for all $i$
4:     $s_1 \leftarrow s_1 - \eta_1 \frac{\partial}{\partial s_1} \sum_i \|\Delta_i\|^2$
5:     $s_2 \leftarrow s_2 - \eta_2 \frac{\partial}{\partial s_2} \sum_i \|\Delta_i\|^2$
6:     project $s_1 \leftarrow \max(s_1, L/M)$, $s_2 \leftarrow \max(s_2, 1)$
7:     **if** $\sum_i \|\Delta_i\|^2 \leq \varepsilon$ **then break**
8:     **end if**
9: **end for**

---

Under standard smoothness and boundedness assumptions on the mapping $(s_1, s_2) \mapsto \partial \mathcal{L}/\partial \theta$, sufficiently small learning rates $\eta_1, \eta_2$ ensure monotone decrease of the objective and convergence to a local minimum.

### D.4 Concluding remarks

- The key analytic object is the scaled phase $\phi_i'(p) = \dfrac{p}{s_1 s_2^{2i/d}}\theta_i$. Ensuring $\phi_i'(L-1)$ lies in the training-phase interval $[0, (M-1)\theta_i]$ for the set of important frequencies $i$ yields inequality equation 23.

- Position interpolation offers a simple, parameter-free method that preserves gradient magnitudes across frequencies in the multiplicative linear approximation. Frequency (base) scaling adds a degree of freedom to prioritize certain frequencies but cannot match all frequencies exactly with a single scalar.

- Practical schemes combine interpolation, frequency-aware scaling, and small calibration fine-tuning on $(s_1, s_2)$ to obtain robust extrapolation; the above derivations give clear, testable predictions for such hybrid methods.

- All theoretical claims are made under a first-order / linearized approximation of attention; nevertheless the derived conditions are both interpretable and useful as initializations or constraints for empirical calibration.

### D.4.1 Strategies for Generalization:

1. Increase M: Increasing the length of the original sequence $M$ allows us to cover a broader range of positions.

2. Adjust Token Positions: Using a method like Position Interpolation (PI), we can adjust the token positions to fall within the range that the model has learned.

3. Increase Base: Using methods such as NTK-aware RoPE, we can increase the base value. Since the positional base is inherently discrete across dimensions, it is only necessary to adjust a subset of the positional encodings correctly to achieve proper adjustment. An advantage of this method is the reuse of previously learned positional bases. For example, bases like $16^{\frac{2}{8}}$ and $256^{\frac{1}{8}}$ can share learned weights, allowing us to generalize to various sequence lengths dynamically.

4. Dimensional Generalization: Similarly, we can extend the generalization mechanism across sequence dimensions. By selectively using parts of the position encodings that fall within the learned range, we can efficiently cover new positions while maintaining computational efficiency.

5. Scaling Numerator and Denominator: We can also consider scaling both the numerator and denominator of position encodings to achieve a more generalized weighting over positional information.

### D.5 Extrapolation capacity limit: model depth

We first analyze potential causes from the perspective of the backward learning process. For the entire training corpus, which we assume to be a sequence of length $L_S$, we partition it into chunks of the pre-training length. Although these chunks are not explicitly modeled as a continuous sequence (i.e., they are not concatenated for end-to-end sequence modeling), the linear superposition of backpropagated gradients across chunks allows the model to capture partial linear relationships between them.

The gradient information propagates backward through each layer, enabling the model to learn most inter-chunk dependencies. Theoretically, between any two adjacent layers, the model can learn information equivalent to a concatenated sequence of two chunks. Consequently, the ultimate extrapolation limit scales linearly with the number of model layers. In the following, we analyze the specific inverse process to illustrate how the above process is implemented:

The gradients of the MLP and the linear layer

**1. Linear Layer**

Given the linear projection:

$$\mathbf{Y} = \mathbf{XW} + \mathbf{b}$$

where: $\mathbf{X} \in \mathbb{R}^{n \times d_{in}}$ is the input, $\mathbf{W} \in \mathbb{R}^{d_{in} \times d_{out}}$ is the weight matrix, $\mathbf{b} \in \mathbb{R}^{d_{out}}$ is the bias.

The gradients during backpropagation (given upstream gradient $\frac{\partial \mathcal{L}}{\partial \mathbf{Y}}$) are:

$$\frac{\partial \mathcal{L}}{\partial \mathbf{W}} = \mathbf{X}^\top \frac{\partial \mathcal{L}}{\partial \mathbf{Y}}, \quad \frac{\partial \mathcal{L}}{\partial \mathbf{b}} = \sum_{i=1}^{n} \left( \frac{\partial \mathcal{L}}{\partial \mathbf{Y}} \right)_i, \quad \frac{\partial \mathcal{L}}{\partial \mathbf{X}} = \frac{\partial \mathcal{L}}{\partial \mathbf{Y}} \mathbf{W}^\top$$

Thus, it is clear:

$$\frac{\partial \mathcal{L}_{\mathrm{cat}(S_m, S_n)}}{\partial \mathbf{W}} = \mathbf{X}_{S_m}{}^\top \frac{\partial \mathcal{L}_{S_m}}{\partial \mathbf{Y}} + \mathbf{X}_{S_n}{}^\top \frac{\partial \mathcal{L}_{S_n}}{\partial \mathbf{Y}}$$

**2. MLP (Multilayer Perceptron)**

With two linear transformations and activation:

$$\mathbf{Z} = \sigma(\mathbf{X}\mathbf{W}_1 + \mathbf{b}_1), \quad \mathbf{Y} = \mathbf{Z}\mathbf{W}_2 + \mathbf{b}_2$$

where $\sigma$ denotes the activation function (e.g., GELU).

**(a) Second Layer Gradients** (identical to linear layer):

$$\frac{\partial \mathcal{L}}{\partial \mathbf{W}_2} = \mathbf{Z}^\top \frac{\partial \mathcal{L}}{\partial \mathbf{Y}}, \quad \frac{\partial \mathcal{L}}{\partial \mathbf{b}_2} = \sum_{i=1}^{n} \left( \frac{\partial \mathcal{L}}{\partial \mathbf{Y}} \right)_i$$

**(b) First Layer Gradients**:

$$\frac{\partial \mathcal{L}}{\partial \mathbf{Z}} = \frac{\partial \mathcal{L}}{\partial \mathbf{Y}} \mathbf{W}_2^\top, \quad \frac{\partial \mathcal{L}}{\partial \mathbf{X}} = \left( \frac{\partial \mathcal{L}}{\partial \mathbf{Z}} \odot \sigma'(\mathbf{X}\mathbf{W}_1 + \mathbf{b}_1) \right) \mathbf{W}_1^\top$$

Here $\odot$ denotes element-wise multiplication, and $\sigma'$ is the derivative of the activation function.

**(c) First Layer Parameter Gradients**:

$$\frac{\partial \mathcal{L}}{\partial \mathbf{W}_1} = \mathbf{X}^\top \left( \frac{\partial \mathcal{L}}{\partial \mathbf{Z}} \odot \sigma'(\mathbf{X}\mathbf{W}_1 + \mathbf{b}_1) \right), \quad \frac{\partial \mathcal{L}}{\partial \mathbf{b}_1} = \sum_{i=1}^{n} \left( \frac{\partial \mathcal{L}}{\partial \mathbf{Z}} \odot \sigma'(\mathbf{X}\mathbf{W}_1 + \mathbf{b}_1) \right)_i$$

From this formula, it can be seen that for sequences whose dimensions are in the Sequence direction, the linear superposition of their gradients is the same as using the inverse gradient of the entire sequence, i.e., these two middle layers completely model the splicing of chunks

The gradients of the Attention layer

Given the attention computation:

$$\mathrm{Attention}(\mathbf{Q}, \mathbf{K}, \mathbf{V}) = \mathrm{softmax}\left( \frac{\mathbf{Q}\mathbf{K}^\top}{\sqrt{d_k}} \right) \mathbf{V}$$

where: $\mathbf{Q}, \mathbf{K} \in \mathbb{R}^{n \times d_k}$ are queries and keys, $\mathbf{V} \in \mathbb{R}^{n \times d_v}$ are values, $n$ is sequence length, $d_k$ is key dimension.

**3. Attention**

1. Output Gradient: Given upstream gradient $\frac{\partial \mathcal{L}}{\partial \mathbf{O}}$ where $\mathbf{O} = \mathrm{Attention}(\mathbf{Q}, \mathbf{K}, \mathbf{V})$:

2. Attention Weights: Let $\mathbf{A} = \frac{\mathbf{Q}\mathbf{K}^\top}{\sqrt{d_k}}$ and $\mathbf{P} = \mathrm{softmax}(\mathbf{A})$

Then:

$$\frac{\partial \mathcal{L}}{\partial \mathbf{P}} = \frac{\partial \mathcal{L}}{\partial \mathbf{O}} \mathbf{V}^\top$$

3. Softmax Gradient:

$$\frac{\partial \mathcal{L}}{\partial \mathbf{A}} = \mathbf{P} \odot \left( \frac{\partial \mathcal{L}}{\partial \mathbf{P}} - \sum_{i=1}^{n} P_{ij} \left( \frac{\partial \mathcal{L}}{\partial \mathbf{P}} \right)_{ij} \right)$$

4. Query/Key Gradients:

$$\frac{\partial \mathcal{L}}{\partial \mathbf{Q}} = \frac{1}{\sqrt{d_k}} \left( \frac{\partial \mathcal{L}}{\partial \mathbf{A}} \right) \mathbf{K}, \quad \frac{\partial \mathcal{L}}{\partial \mathbf{K}} = \frac{1}{\sqrt{d_k}} \left( \frac{\partial \mathcal{L}}{\partial \mathbf{A}} \right)^{\top} \mathbf{Q}$$

5. Value Gradient:

$$\frac{\partial \mathcal{L}}{\partial \mathbf{V}} = \mathbf{P}^{\top} \frac{\partial \mathcal{L}}{\partial \mathbf{O}}$$

For simplicity, we analyze the last two layers to learn the gradient information of one chunk first and then perform Attention on the latter chunk, with the error of using two chunks spliced together for Attention to illustrate that the model is able to learn a part of the information of the spliced Attention, thus realizing a certain degree of extrapolation ability.

$$\Delta_o = \text{Attention}(\mathbf{Q}_{S_n}, \mathbf{K}_{\text{cat}(S_m, S_n)}, \mathbf{V}_{\text{cat}(S_m, S_n)}) - \text{Attention}(\mathbf{Q}_{S_n}^{\mathbf{new}}, \mathbf{K}_{S_n}^{\mathbf{new}}, \mathbf{V}_{S_n}^{\mathbf{new}})$$

$$\Delta_o = \text{softmax} \left( \frac{\mathbf{Q}_{S_n} \mathbf{K}_{S_m}^{\top}}{\sqrt{d_k}} \right) \mathbf{V}_{S_m} + \text{softmax} \left( \frac{\mathbf{Q}_{S_n} \mathbf{K}_{S_n}^{\top}}{\sqrt{d_k}} \right) \mathbf{V}_{S_n} - \left( \text{softmax} \left( \frac{\mathbf{Q}_{S_n}^{\mathbf{new}} \mathbf{K}_{S_n}^{\mathbf{new}\top}}{\sqrt{d_k}} \right) \mathbf{V}_{S_n}^{\mathbf{new}} \right)$$

where:

$$\mathbf{Q}_{S_n}^{\mathbf{new}} = \mathbf{Q}_{S_n} - \frac{\partial \mathcal{L}}{\partial \mathbf{Q}_{S_m}}, \mathbf{K}_{S_n}^{\mathbf{new}} = \mathbf{K}_{S_n} - \frac{\partial \mathcal{L}}{\partial \mathbf{K}_{S_m}}, \mathbf{V}_{S_n}^{\mathbf{new}} = \mathbf{V}_{S_n} - \frac{\partial \mathcal{L}}{\partial \mathbf{V}_{S_m}},$$

For ease of analysis we freeze the gradients of Q and V (gradients with Q and V will have smaller errors) and observe only the output after learning K.

$$\mathbf{O}_{S_n}^{\mathbf{new}} = \left( \text{softmax} \left( \frac{\mathbf{Q}_{S_n} \left( \mathbf{K}_{S_n} - \frac{1}{\sqrt{d_k}} \left( \frac{\partial \mathcal{L}}{\partial \mathbf{A}} \right)^{\top} \mathbf{Q}_{S_m} \right)^{\top}}{\sqrt{d_k}} \right) \mathbf{V}_{S_n} \right)$$

Observing only the propagation of information, it can be seen that the learned Attention is able to approximate the Attention with Q with the information of the previous sequence $S_m$, thus completing a portion of the complete Attention information, and then each chunk, with each layer of attention, is able to realize an approximate Attention of the L-fold sequence, thus realizing a certain amount of extrapolation. Finally more detailed and rigorous proofs we leave for future work to investigate.

### D.5.1 Other Extrapolation Error Analysis:

To analyze the error during extrapolation, we assume a linear relationship between different segments, where each part is responsible for the corresponding position transformation. Specifically, the part of the gradient involving $P_c$ is responsible for handling the ground-truth $P_c$. Let the positional difference between seen and unseen sequences be:

$$Q_Z - Q_L = X W_{QZ} - X \left( W_Q + \frac{\partial \mathcal{L}}{\partial W_Q} \right)$$

Now, consider a fine-tuning setup where we train on sequences of length $M$. For the first $M$ tokens, the model performs without loss. However, for tokens beyond $M$, the extrapolation requires additional adjustments. For the first $M$ tokens, the gradient is:

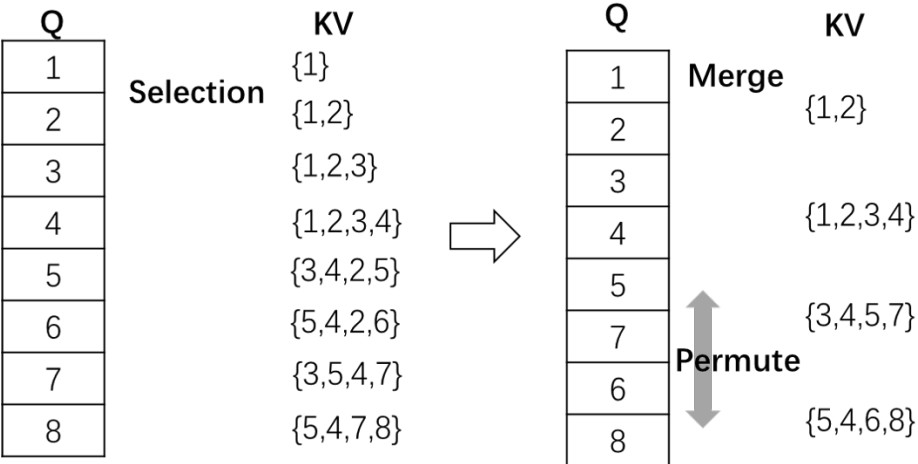

Figure 4: **Overview of Merging and Selection Attention Mechanism (MS Attention).** The MS Attention mechanism involves two main steps. In the first step, the QKV tensors are split into regions, and a single token is used to represent each region. Subsequently, the regional representatives are used to compute dot products or other similarity measures to select the most relevant KV regions for each Q region. For example, Q regions 5, 6, 7 and 8 select KV regions $\{3, 4, 2, 5\}$, $\{5, 4, 2, 6\}$, $\{3, 5, 4, 7\}$ and $\{5, 4, 7, 8\}$ respectively. In the second step, for each Q region, tokens are merged with their adjacent or related tokens after permuting. The union of the selected KV regions is taken, and the top-n regions are chosen. For example, combine Q regions 6 and 8, along with their selected KV regions, resulting in $\{5, 4, 2, 7, 6, 8\}$. To ensure tensor consistency, we select the top $k$ regions from the merged set. If $k = 4$, the final selection is $\{5, 4, 6, 8\}$. The reason for retaining 6 and 8 is because by default we believe that the local region is important and must be preserved, and that the local region does not perform scoring calculations. Finally, each merged Q region performs self-attention with its selected relevant KV regions.

$$X_{:M} W_{QZ} = X_{:M} \left( W_Q + \frac{\partial \mathcal{L}}{\partial W_Q} \right) + L(V, K)$$

For the tokens beyond $M$, the extrapolation error is:

$$(X_{M:} W_{QZ}) \odot P_c - (X_{M:}(W_Q + \frac{\partial \mathcal{L}}{\partial W_Q})) \odot P_c$$

Letting it generalize continuously over a fixed length enables the continuous reduction of the above error L, thus reducing the extrapolation of the final position information.

# E    Analysis of Attention with Correlation-Aware Selection and Merging:

## E.1    Approximation and Convergence of Our MS Attention to Various Efficient Attention Methods

Our proposed Select-Merge Attention (MS Attention) can approximate and converge to the optimal solutions of many efficient Attention mechanisms that utilize a subset of key-value (KV) pairs. Below, we describe the computation process using the notations introduced in the algorithm description:

Selection:
$$A_s = Q'_s K'^T_s, \quad \text{Idx} = \text{topkIndex}(A_s)$$

Merge:
$$Q_s = \text{merge}_q(Q'_s), \quad \text{Idx} = \text{filter}(\text{merge}_q(\text{Idx})), \quad KV_s = \text{Select}(KV, \text{Idx})$$

---

**Algorithm 2** Correlation-based Selection & Merging for Sparse Attention

---

**Require:** $Q, K, V \in \mathbb{R}^{B \times H \times N \times D}$          ▷ batch $B$, heads $H$, tokens $N$, dim $D$
**Require:** segment sizes $s_q, s_k$; number of $Q$-segments $n_s \leftarrow \lceil N/s_q \rceil$; *topk*; merge factor $m$ (divisor of $n_s$)
**Require:** compress function compress$(\cdot)$ (e.g. mean / learned semantic token)
**Require:** similarity function sim$(\cdot, \cdot)$ (e.g. dot-product)
**Require:** mask function mask$(\cdot)$ (to prevent leakage); optional final select length *top_final*
**Ensure:** Output $\in \mathbb{R}^{B \times H \times N \times D}$ (approximate attention output)
  1: **Segmentation:**
  2: Split $Q$ into $Q_s \in \mathbb{R}^{B \times H \times n_s \times s_q \times D}$ by segments          ▷ last segment padded if needed
  3: Split $K, V$ into $K_s \in \mathbb{R}^{B \times H \times n_k \times s_k \times D}$, $V_s$ similarly
  4: **Compress region representatives:**
  5: $Q'_s \leftarrow$ compress$(Q_s)$,   $Q'_s \in \mathbb{R}^{B \times H \times n_s \times D}$          ▷ one token per $Q$-segment
  6: $K'_s \leftarrow$ compress$(K_s)$,   $K'_s \in \mathbb{R}^{B \times H \times n_k \times D}$
  7: **Compute relevance / selection:**
  8: $S \leftarrow$ sim$(Q'_s, K'_s)$          ▷ $S \in \mathbb{R}^{B \times H \times n_s \times n_k}$
  9: $S \leftarrow$ mask$(S)$          ▷ mask out invalid / future / leakage entries
10: $selectindx \leftarrow$ topk_indices$(S, k = topk)$          ▷ $(B, H, n_s, topk)$: top-$k$ $K_s$ indices per $Q_s$
11: **(Optional) Permutation for sharing/clustered $KV$ (pretrain/infer):**
12: Compute permutation $\pi$ from $S$ (group similar $Q_s$)          ▷ optional, improves $KV$ sharing
13: Apply $\pi$ to $Q_s$ and $selectindx$ (and record inverse $\pi^{-1}$)
14: **Merging $Q$ segments:**
15: $n_{ms} \leftarrow n_s/m$          ▷ number of merged $Q$ regions
16: Reshape / merge $Q_s$ into $Q_{ms} \in \mathbb{R}^{B \times H \times n_{ms} \times (m \cdot s_q) \times D}$ by concatenating every $m$ adjacent $Q_s$
17: Optionally compress merged $Q_{ms}$ to one representative per merged region (if desired)
18: **Transform and merge selected indices:**
19: Reshape $selectindx$ from $(B, H, n_s, topk)$ to $(B, H, n_{ms}, m, topk)$ by grouping every $m$ adjacent $Q_s$ rows
20: Permute to $(B, H, n_{ms}, topk, m)$ and then reshape to $(B, H, n_{ms}, topk \cdot m)$ ▷ now each merged $Q$ region has $topk \cdot m$ candidate $K_s$
21: For each merged row do:   $mselectindx \leftarrow$ unique_preserve_order(candidates) ▷ remove duplicates while keeping relevance order
22: $qmselectindx \leftarrow$ top_n$(mselectindx, n = top\_final)$      ▷ final top-$n$ $K_s$ groups per merged $Q$ region; shape $(B, H, n_{ms}, top\_final)$
23: **Gather relevant $K/V$ segments:**
24: For each merged $Q$ region, gather the corresponding $K_s$ and $V_s$ regions indexed by $qmselectindx$ and similarly $V^g_{ms}$:
$$K^g_{ms} \in \mathbb{R}^{B \times H \times n_{ms} \times top\_final \times s_k \times D}$$
25: **Compute attention on merged blocks:**
26: **for** each merged region index $r = 1 \ldots n_{ms}$ **do**
27:      Flatten $Q_{ms}[..., r]$ to queries of shape $(B, H, qlen = m \cdot s_q, D)$
28:      Flatten $K^g_{ms}[..., r]$ and $V^g_{ms}[..., r]$ across selected segments to $(B, H, klen, D)$ where $klen = top\_final \cdot s_k$
29:      Compute local attention outputs:

$$O_{ms}[..., r] \leftarrow \text{Attention}\big(Q_{ms}[..., r], \ K^g_{ms}[..., r], \ V^g_{ms}[..., r]\big)$$

30: **end for**
31: **Unmerge and scatter back to token positions:**
32: Split each $O_{ms}[..., r]$ back to the $m$ original $Q_s$ regions and to individual tokens, then reorder according to $\pi^{-1}$ if permutation was applied and get $\hat{O} \in \mathbb{R}^{B \times H \times N \times D}$.
33: Attention computation process can be fully parallelised.
34: **return** $\hat{O}$

---

Final Attention:

$$O = \text{Attention}(Q_s, K_s, V_s)$$

### E.1.1 Landmark Attention

Landmark Attention adjusts the attention scores by incorporating landmark tokens, with the output expressed as:

$$O = (\text{softmax}(QK^T) \cdot \text{repeat}(\text{softmax}(QG^T), \text{blocksize}, \text{dim} = -1))V$$

When setting the split size of $Q$ to 1 and the split size of $K$ to blocksize, our MS Attention also calculates $\text{softmax}(QK_s^T)$ and $\text{softmax}(QK_s'^T)$, which fully enables the adjustment of attention scores using semantic tokens. This operation can be approximated under other settings as well.

Alternatively, if the score adjustment is performed without directly multiplying $\text{softmax}(QK_s'^T)$, treating semantic tokens as regular KV tokens during Attention computation, the output can be expressed as:

$$\frac{\text{sumexp}(QK_s^T) \times \text{softmax}(QK_s^T)V_s}{\text{sumexp}(QK_s^T) + \text{sumexp}(QK_s'^T)} + \frac{\text{sumexp}(QK_s'^T) \times \text{softmax}(QK_s'^T)V_s'}{\text{sumexp}(QK_s^T) + \text{sumexp}(QK_s'^T)}$$

where $V_s'$ is a linear combination of $V_s$ or $V$. This approach can also directly adjust the attention coefficient of $V_s$ in the output, allowing convergence to the optimal solution based on the task.

### E.1.2 BiFormer

The BiFormer computation process is as follows:

$$A_r = Q_r(K_r)^T, \quad I_r = \text{topkIndex}(A_r), \quad KV_g = \text{Select}(KV, I_r)$$

$$O = \text{Attention}(Q, K_g, V_g)$$

When the Merge size is set to 1, our method fully degenerates to BiFormer. To achieve lower complexity, BiFormer requires setting the initial region $Q_r, K_r$ relatively large and then compressing it as a region representative. This approach reduces the accuracy of selection.

When merge size is greater than 1, our method performs more fine-grained partitioning and selection. If BiFormer's selection is optimal, our algorithm can also converge to this optimal selection. Overall, compared to BiFormer, our method has a larger convergence space, and under the same fine partitioning and selection, our method has relatively lower computational complexity.

### E.1.3 Routing Attention Methods

For the Routing Transformer, the following update rule is applied:

$$\mu \leftarrow \lambda\mu + \frac{(1-\lambda)}{2} \sum_{i:\mu(Q_i)=\mu} Q_i + \frac{(1-\lambda)}{2} \sum_{j:\mu(K_j)=\mu} K_j$$

This can be rewritten as:

$$\mu \leftarrow \lambda\mu + \frac{(1-\lambda)}{2}\text{argmax}_{qu}(\mu Q^T)Q + \frac{(1-\lambda)}{2}\text{argmax}_{qu}(\mu K^T)K$$

where

$$\text{argmax}_{qu}(\cdot) = \begin{cases} 1 & \text{argmax}(\cdot, \text{dim} = -2) \\ 0 & \text{otherwise} \end{cases}$$

$$\text{Idx}_Q = \text{topkIndex}(\mu Q^T), \quad \text{Idx}_K = \text{topkIndex}(\mu K^T)$$

Using the triangle inequality of the metric, the above approximation can select $Q$-related KV tokens with:

$$\text{Idx} = \text{topkIndex}(Q\mu^T \mu K^T)$$

Our semantic tokens in each Attention step also cluster similarly:

$$m_s = \text{Softmax}(m_s K_s^T) K_s$$

or

$$m_s = m_s + \text{softmax}(Q_s K_s^T) V_s$$

Our selection process can be written as:

$$\text{Idx} = \text{topkIndex}(m_s W_Q W_K^T m_s^T)$$

where $m_s$ is the regional semantic token of $X_s$ (e.g., average or $m_s = \text{Softmax}(m_s K_s^T) K_s$), equivalent to further clustering.

Therefore, during selection, the cluster center can be used to represent the tokens within the relevant cluster, similar to the Routing Transformer. Our method, by only using the cluster center for selection, introduces minimal quantization error. However, because the selection quantity is sufficient, the loss is negligible. In contrast, the Routing Transformer loses some related $Q$ and $K$ tokens to ensure regular shape, introducing non-negligible error.

### E.1.4 Swin Transformer

The Swin Transformer utilizes local Attention and shifted local Attention. Our selection mechanism can completely converge to Swin Transformer when it is optimal.

- Local Attention:

$$\text{softmax}(Q_{i:j} K_{i:j}^T) V_{i:j}$$

When this is the optimal solution, our selection mechanism will automatically select tokens within the local region.

- Shifted Local Attention:

$$\text{softmax}(Q_{i:j} K_{i+r:j+r}^T) V_{i+r:j+r}$$

When this is optimal, our selection mechanism will automatically select tokens within the corresponding region ($KV_{i+r:j+r}$), where $r$ is the cyclic shift offset.

In a similar manner, our method can approximate or cover many variants of the above Transformers.

To compare our method with these efficient Transformers, we conducted experiments on the PG19 datasets, as shown in the tables 30.

Table 30: Perplexity (PPL) and Training Memory Consumption (MB) on PG19. The 20186/22496 means using flashatten or not. The $1 \times 8192$ means $bsz \times seqlen$.

| Model | PPL (PG19) | $1 \times 8192$ | $1 \times 4096$ | $1 \times 2048$ |
|---|---|---|---|---|
| **MS Attention-60M** | 13.9 | - | - | - |
| **MS Attention-200M** | 9.32 | 20186/22496 | 13152/14300 | 9976/10346 |
| **Landmark Attention-200M** | 14.55 | >40960 (OOM) | >40960 (OOM) | 17938 |

On the PG19 dataset, our method outperforms Landmark Attention in terms of both performance and efficiency, improving PPL by 4. The efficiency advantage becomes more pronounced as the input length increases.

### E.2 Detailed Parameter Settings

Our algorithm can encompass the majority of Q-attention to KV-subset methods by adjusting parameters such as the QKV segment size, the number of selected top-K high-relevance KV regions, and the number of merged Q regions. Below is a detailed discussion on the selection of these hyperparameters:

### E.2.1 QKV Segment Size Selection:

The QKV segment size is crucial and typically ranges from 8 to 128. This parameter must be chosen by balancing computational complexity and performance. In future work, we plan to incorporate Triton operators in the QK routing step to achieve linear spatial complexity, thereby mitigating the current limitations:

- (1) During the selection and routing process, a semantic token represents each region, and the relevance between Q and K semantic tokens is measured. KV tokens related to the Q region are then selected based on this relevance. A larger QK region size results in a coarser semantic representation, leading to less precise KV token selection. To minimize this loss, more KV tokens must be selected in the second step, thereby reducing information loss due to coarse semantics. Thus, smaller segment sizes are preferred, though incorporating Triton operators in the QK routing step is anticipated to provide linear spatial complexity.

- (2) Alternatively, a larger segment size can be set, and more KV tokens can be selected in the second step. This approach may increase algorithmic complexity as more KV tokens are likely to be used for interpolation within the same Q region. However, this issue can be alleviated through the third step of merging, where the selected KV tokens can be further merged and selected according to relevance, filtering out irrelevant information.

- (3) Additionally, the segment size of Q regions is independent of the segment size of KV regions. A relationship between them should only be established when specific tasks and complexity constraints require it.

### E.2.2 Top-K High-Relevance KV Region Selection:

The number of selected top-K high-relevance KV regions is generally determined by factors such as the model's pre-training length and the amount of task-specific data. It may also need to be adjusted based on the granularity of the segmentation from the first step. Current large model training methods suggest that selecting 1K-8K KV tokens for interpolation is robust:

- (1) The above represents a general approach, while scenario-specific selection yields higher performance and efficiency. For example, if the task's data volume is small, a relatively small number of high-relevance regions can be selected to maximize overfitting and memorize the critical data. Conversely, as the data volume increases, more KV tokens may be required for autoregressive prediction of the next token, necessitating the selection of more high-relevance regions.

- (2) In addition to task-based selection, the granularity of segmentation from the first step must also be considered. If segmentation precision is insufficient, more KV tokens should be selected to prevent the loss of critical information. Subsequent merging can then be used to jointly select high-relevance information, improving space and time complexity by sharing KV tokens across Q regions.

- (3) For length fine-tuning, the model's parameters have adapted to the interpolation degree of the pre-training length, which uses a specific number of KV tokens. Hence, the same magnitude of KV tokens must be selected during fine-tuning to avoid overfitting.

This parameter is highly flexible and can be adapted to various scenarios, often requiring some experience for optimal settings. Future work may involve incorporating a loss function to control the selection space, allowing the algorithm to automatically select the appropriate size based on the loss.

### E.2.3 Merging Q Regions:

The number of merged Q regions generally depends on the segmentation precision and space complexity. In scenarios where space complexity is not a major concern, sizes ranging from $\frac{topk}{4C_Q}$ to $\frac{topk}{C_Q}$ can be chosen. The number of merged KV tokens is typically selected from a range of topk to 2topk KV tokens. Generally, sharing high-relevance KV regions across multiple Q regions results in lower space complexity. Moreover, during the merging process, high-relevance KV regions are selected with flexibility, enhancing performance through methods such as:

- a. Simple unique and sort operations for selection.

- b. Unique operations followed by a secondary segmentation based on high-relevance scores, with allocation and selection according to region clustering.

### E.2.4 Algorithm Complexity Analysis

(1) Let the segmentation sizes for the Q and KV regions be $C_Q$ and $C_K$, respectively. The time and space complexity of routing Q and K using dot products is $O\left(\frac{N^2}{C_Q C_K}\right)$.

(2) In the second step, the selection of high-relevance regions, denoted by $C_S$, results in space complexity of $O\left(\frac{N}{C_Q}C_S\right)$ due to the storage of necessary indices.

(3) The merging step, with a merge size of $C_M$, involves combining the selected indices. Filtering algorithms can be introduced during merging; in this work, we employ unique and high-score selection. The space complexity for storing KV tokens after selection and merging is $O\left(\frac{N^2}{C_Q C_M}C_S\right)$.

(4) The final attention operation has a time complexity of $O(2\frac{N}{C_Q C_M}C_Q C_M C_S) = O(NC_S)$. Hence, only the segmentation size and selection quantity affect the algorithm's complexity.

Without merging, the selected KV tokens have a complexity of $O\left(\frac{N^2}{C_Q}C_S\right)$, whereas with merging, the complexity is $O\left(\frac{N^2}{C_Q C_M}C_S\right)$. The reduction in complexity due to merging is because the selection process is only carried out after merging, storing only the selected indices before that.

### E.3 Reduced LongLora

We propose a method that selectively fine-tunes only the $W_k$ and $W_o$ weights, achieving results nearly identical to fine-tuning the entire attention mechanism. Specifically, since $QK^T = XW_q W_k^T X^T$, updating only $W_k$ yields $W_q W_k^T$, effectively replicating the effect of simultaneously updating $W_k$ and $W_q$. This approach is particularly effective when $W_q$ is full rank. Similarly, fine-tuning $W_o$ follows the same rationale, as the final output is a linear mapping of $W_v W_o$. Since only the $Q$ and $K$ tokens undergo positional encoding, the weights to be fine-tuned are limited to $W_Q$ and $W_K$. Based on the above analysis, it suffices to only fine-tune $W_Q$ or $W_K$. In this work, for the fine-tuning of LLaMA, we adopt a low-rank fine-tuning of $W_Q$ and $W_K$, while for Mistral, we apply low-rank learning of $W_Q$ for sequence length extension.

Additionally, since query tokens can be considered well-fitted through extensive training, learning the linear mapping for their corresponding key tokens is reasonable. Furthermore, as the attention mechanism becomes heterogeneous, updating the final classification head is a direct approach. However, considering the large number of parameters, altering the initial embeddingAn et al. (2022) to achieve a certain degree of equivalence is also a feasible consideration.

### E.3.1 The strategy for incorporating positional information during pre-training

Our algorithm leverages semantic token routing, where each semantic token encapsulates contextual information from its surrounding tokens, similar to a sliding window mechanism in convolution. Since different tokens are surrounded by distinct contexts, the resulting semantic tokens are unique, leading to different routed $KV$ tokens and interpolated variables. As a result, we do not assign positional information during the selection process. Instead, after the selection process, RoPE positional encoding is applied to the $K$ and $Q$ tokens to enhance performance.

