# OpenReview forum: "An Efficient Framework for Length Extension via Dynamically Growing Positional Embedding and Routing Attention"
_TMLR — Under review for TMLR_

### Review · Reviewer_zTtR · 2026-03-11

**Summary Of Contributions:**

The paper studies long-context extension for pretrained LLMs and proposes a two-part framework: (1) CRG-NTK, a RoPE-based positional-encoding scheme that combines cyclic/random truncation with dynamically growing scaling factors to improve extrapolation, and (2) MS Attention, a routing-based sparse attention mechanism that selects and merges relevant regions to reduce computation and suppress noise in long sequences. The paper reports strong extrapolation results on passkey retrieval, long-context perplexity, RULER, and LongBench, and argues that its method can achieve similar extension quality with substantially less fine-tuning compute than prior approaches such as LongRoPE and LongLoRA.

The main strengths are that the paper tackles a practically important problem, combines positional scaling with an efficiency-oriented attention mechanism, and presents a fairly broad empirical section across multiple models and tasks. The main weaknesses are that the contribution is quite bundled, the theoretical claims—especially the proposed extrapolation limit scaling with model depth—are not convincingly established, and several empirical comparisons are not sufficiently controlled to support the broader claims as strongly as written. In particular, the method mixes CRG-NTK, MS Attention, LoRA choices, and selective parameter tuning, making it difficult to isolate where the gains come from.

**Audience:**

Yes

**Audience Explanation:**

Yes. Efficient long-context extension for pretrained LLMs is an active and important topic, and many TMLR readers would likely be interested in practical methods that reduce the cost of context-window extension. The proposed combination of dynamic RoPE scaling and routing-based sparse attention is relevant to researchers working on long-context modeling, positional encoding, efficient attention, and post-training adaptation of language models. Even if I am not fully convinced by all of the paper’s claims, the empirical findings and design choices are still of potential interest to that audience.

**Broader Impact Concerns:**

None. There is no significant broader impact concerns related to this paper.

**Claims And Evidence:**

No

**Claims Explanation:**

The paper does provide a nontrivial amount of empirical evidence: passkey retrieval, PG19/proofpile perplexity, LongBench, RULER, ablations on MS Attention settings, and some training-time comparisons are all included. This is enough to suggest that the proposed recipe can work in practice. However, I do not think the evidence is sufficiently clear or convincing for several of the stronger claims.

* The paper’s main gains come from a package of changes rather than a cleanly isolated idea. The submission combines CRG-NTK positional scaling, random/cyclic truncation, dynamic growth, MS Attention, LoRA-based fine-tuning, and selective parameter updates. Because of this, it is hard to tell how much of the improvement comes from the positional method itself versus the routing attention and fine-tuning setup.

* Some of the central conceptual claims appear speculative relative to the evidence. In particular, the claim that the extrapolation limit is approximately bounded by model depth is interesting, but the support given is mostly empirical trend observation from a small set of models rather than a convincing theoretical or systematically controlled demonstration.

* The comparison methodology is not fully controlled. In several places the paper compares its 16K fine-tuning setup against prior methods that use different fine-tuning lengths, search budgets, and architectures. The resource-efficiency claim may be directionally true, but it is difficult to interpret it as a rigorous apples-to-apples comparison. Likewise, some evaluations compare MS Attention at test time against baselines using different attention mechanisms or training recipes, which makes the practical advantage harder to disentangle.

The paper’s presentation makes some of the conclusions seem broader than what is directly demonstrated. Overall, I believe the experiments are promising, but the current evidence falls short of fully supporting the strongest accuracy, extrapolation, and efficiency claims in a clear and convincing way.

**Requested Changes:**

1. Provide cleaner component-wise ablations that isolate the contribution of CRG-NTK from MS Attention, and further separate cyclic truncation, random shift, and dynamic growth. Right now the paper evaluates the full recipe more than the individual ideas, which makes attribution difficult. Substantially strengthen the evidence for the claim that extrapolation scales with model depth. At minimum, this needs either a more rigorous theoretical argument with clearly stated assumptions or a broader, better-controlled empirical study across models where depth is varied more systematically.

2. Improve the fairness of empirical comparisons. For the main efficiency and quality claims, compare against strong baselines under matched or at least clearly normalized training budgets, context lengths, and attention settings. The current comparisons to LongRoPE/LongLoRA are suggestive, but not rigorous enough for the strength of the conclusions. Clarify the exact test-time setting in every main table: whether evaluation uses Full Attention or MS Attention, whether the baseline uses the same attention type, and whether inference-time efficiency claims are measured under comparable implementations.

3. Tone down or qualify claims that are currently stated too broadly, especially around “arbitrary length” extrapolation, the generality of the depth-based extrapolation limit, and the magnitude of the practical efficiency advantage.

---

> ### Author Response · Authors · 2026-05-13
> **Provide cleaner component-wise ablations that isolate the contribution of CRG-NTK from MS Attention, and further separate cyclic truncation, random shift, and dynamic growth. Right now the paper evaluates the full recipe more than the individual ideas, which makes attribution difficult. Substantially strengthen the evidence for the claim that extrapolation scales with model depth.**
>
> Thank you very much for your valuable comments. In particular, we appreciate your suggestion to provide cleaner component‑wise ablations that isolate the contribution of CRG‑NTK from MS Attention, and further separate cyclic truncation, random shift, and dynamic growth. We fully agree that the effect of each component should be clearly presented for the reader’s understanding and ease of use. In our original paper we already included many ablation experiments for each component, but they were scattered across different experiments and may have been overlooked. Following your suggestion, we have now consolidated them and added a few more experiments to make the picture more complete. The results are shown in the table below.
>
> | Method | PG19‑PPL ↓ (Finetuning 16K, test: FA) | | | | Passkey Retrieval ↑ | | | | |
> |:---|---:|---:|---:|---:|---:|---:|---:|---:|---:|
> | | 32K | 64K | 128K | 256K | 32K | 64K | 128K | 256K | 512K |
> | **PLA‑POS** LoRA + MS Attn + CRG‑NTK | **7.25** | **7.29** | **7.43** | **8.36** | **1.0** | **1.0** | **1.0** | **1.0** | **1.0** |
> | w/o LoRA: MS Attn + CRG‑NTK | 7.18 | 7.26 | 7.42 | 8.39 | 1.0 | 1.0 | 1.0 | 1.0 | 1.0 |
> | w/o MS Attn: LoRA + CRG‑NTK | 7.10 | 7.23 | 7.38 | 8.34 | 1.0 | 1.0 | 1.0 | 1.0 | 1.0 |
> | w/o CRG‑NTK: LoRA + MS Attn | >10³ | >10³ | >10³ | >10³ | 0.1 | 0.0 | 0.0 | 0.0 | 0.0 |
> | LoRA + MS Attn (test: MS) | 7.02 | 7.14 | 8.57 | 15.14 | 0.3 | 0.1 | 0.0 | 0.0 | 0.0 |
> | LoRA + NTK | 7.69 | 8.08 | 8.42 | 9.32 | 1.0 | 1.0 | 0.9 | 0.6 | 0.1 |
> | LoRA + MS Attn + NTK | 7.87 | 8.19 | 8.46 | 9.20 | 1.0 | 1.0 | 1.0 | 0.7 | 0.2 |
> | LoRA + NTK + random shift | 7.91 | 7.96 | 8.38 | 8.78 | 1.0 | 1.0 | 1.0 | 0.9 | 0.2 |
> | LoRA + NTK + random shift + cyclic truncation | 7.92 | 7.97 | 8.36 | 8.68 | 1.0 | 1.0 | 1.0 | 1.0 | 0.3 |
>
> The table shows that if full attention is used for testing, the extrapolation ability comes almost entirely from CRG‑NTK, which is capable of achieving extremely high extrapolation ratios. NTK alone also achieves reasonable extrapolation, but with some loss on the retrieval task. Adding random shift and cyclic truncation further improves performance at longer lengths, especially for passkey retrieval. The most important factor, however, is dynamic growth, which greatly boosts passkey retrieval performance and extrapolation length. When testing with MS Attention, there is also some extrapolation ability because selecting a fixed‑size subset for autoregressive inference can mitigate noise and achieve better performance and longer extrapolation than full attention. However, due to the very limited fine‑tuning budget, using MS Attention for testing does not yield exact results on the retrieval task.

---

> > ### Author Response · Authors · 2026-05-13
> > **At minimum, this needs either a more rigorous theoretical argument with clearly stated assumptions or a broader, better-controlled empirical study across models where depth is varied more systematically.**
> >
> > Regarding the claim that extrapolation scales with model depth, we provided an analysis in Appendix D.10 of the original paper. From the perspective of gradient information, we argue that the product of depth and training length provides information that overlaps with directly learning the length. A rigorous proof would require substantial space. Since the main focus of our paper is to develop an efficient method for extending the context window of LLMs, and the depth‑dependent extrapolation is only a phenomenon we wish to highlight, we do not intend to provide a complete theoretical proof. Instead, we have added some small experiments to provide supporting evidence. We trained small models of 2, 4, 8, 16, and 32 layers with CRG‑NTK, RoPE, and no positional encoding, using a training window of 4096 and 2.5B tokens. The initial base of both CRG‑NTK and RoPE was set to 10000, and for CRG‑NTK we doubled the base every 0.25B tokens. We then tested extrapolation from 2K to 128K. For testing, CRG uses the final base multiplied by 1024, while CRGs uses base × 1024 × (L / 4096). The results are shown below.
> >
> > | Model / PG19 (PPL) | 2048 | 4096 | 8192 | 16384 | 32768 | 65536 | 131072 |
> > |--------------------|------|------|------|-------|-------|-------|--------|
> > | Layers2‑CRG        | 94.1987 | 105.3854 | 131.5294 | 175.0876 | 196.0545 | 243.5528 | 240.1827 |
> > | Layers4‑CRG        | 86.3804 | 89.7223 | 104.8467 | 135.5123 | 165.6186 | 209.4230 | 210.8753 |
> > | Layers8‑CRG        | 67.5048 | 72.4100 | 88.0006 | 112.3487 | 150.3726 | 209.4400 | 207.4952 |
> > | Layers16‑CRG       | 57.0404 | 62.2363 | 73.0349 | 93.3110 | 112.0631 | 141.0800 | 136.6930 |
> > | Layers32‑CRG       | 46.9055 | 50.0708 | 57.2989 | 70.9097 | 88.0831 | 116.4049 | 112.2660 |
> >
> > | Model / PG19 (PPL) | 2048 | 4096 | 8192 | 16384 | 32768 | 65536 | 131072 |
> > |--------------------|------|------|------|-------|-------|-------|--------|
> > | Layers2‑NoPE       | 173.9474 | 172.8089 | 180.8761 | 191.5813 | 184.1269 | 192.5859 | 191.4170 |
> > | Layers4‑NoPE       | 116.5275 | 121.8906 | 141.7939 | 172.9171 | 189.3198 | 208.7324 | 202.5107 |
> > | Layers8‑NoPE       | 54.3092 | 56.5076 | 69.8976 | 125.8583 | 248.7006 | 673.8378 | 1567.4213 |
> > | Layers16‑NoPE      | 45.3730 | 57.7778 | 64.4023 | 143.7122 | 325.3820 | 573.8408 | 795.3875 |
> > | Layers32‑NoPE      | 63.7127 | 65.9816 | 101.2090 | 202.9836 | 333.3065 | 463.0249 | 681.2415 |
> >
> > | Model / PG19 (PPL) | 2048 | 4096 | 8192 | 16384 | 32768 | 65536 | 131072 |
> > |--------------------|------|------|------|-------|-------|-------|--------|
> > | Layers2‑RoPE       | 88.0872 | 91.3877 | 124.7565 | 215.7571 | 244.0810 | 266.5466 | 270.4461 |
> > | Layers4‑RoPE       | 59.5189 | 62.0980 | 93.9253 | 172.2161 | 229.2213 | 260.3127 | 267.0177 |
> > | Layers8‑RoPE       | 54.7436 | 57.4432 | 88.5246 | 164.9471 | 234.6571 | 266.6695 | 277.2006 |
> > | Layers16‑RoPE      | 49.6317 | 50.8603 | 67.2303 | 133.1615 | 204.0648 | 251.1964 | 259.1832 |
> >
> > | Model / PG19 (PPL) | 2048 | 4096 | 8192 | 16384 | 32768 | 65536 | 131072 |
> > |--------------------|------|------|------|-------|-------|-------|--------|
> > | 8‑CRGs             | 67.5048 | 72.3761 | 83.6303 | 121.7260 | 139.9888 | 180.2314 | 163.3717 |
> > | 16‑CRGs            | 57.0404 | 62.2363 | 69.8757 | 88.6216 | 97.7279 | 120.3819 | 107.8512 |
> > | 32‑CRGs            | 46.9055 | 50.0787 | 54.3343 | 64.3565 | 77.2488 | 97.4854 | 86.4111 |
> >
> > The conclusions are as follows. The performance of the Layers{m}‑CRG model at length L is similar to that of the Layers{2m}‑CRG model at length 2L. Moreover, deeper layers (or larger parameter count) lead to better initial performance. This relationship between extrapolation ability and depth or parameter count can serve as a foundation for future scaling laws of extrapolation.

---

> > > ### Author Response · Authors · 2026-05-13
> > >
> > > 2. Improve the fairness of empirical comparisons. For the main efficiency and quality claims, compare against strong baselines under matched or at least clearly normalized training budgets, context lengths, and attention settings. The current comparisons to LongRoPE/LongLoRA are suggestive, but not rigorous enough for the strength of the conclusions. Clarify the exact test‑time setting in every main table: whether evaluation uses Full Attention or MS Attention, whether the baseline uses the same attention type, and whether inference‑time efficiency claims are measured under comparable implementations.
> > >
> > > Our comparisons are all conducted under fair settings, and in fact we deliberately made the baselines as strong as possible. First, our fine‑tuning budget is far smaller than that of LongRoPE, LongLoRA, and YaRN. We fine‑tune with 16K and 64K lengths using 2 A100 GPUs. Each batch size is 1 (because only the learnable matrices of LoRA with rank 16 for q and k receive the information, so a large batch size is not necessary). We fine‑tune for 10K steps, resulting in a total of 320M and 1.25B tokens. In contrast, LongRoPE uses 128K and 256K lengths with a global batch size of 64 for 1K steps, requiring 8B and 16B tokens. However, this is only a small part of LongRoPE’s cost, as LongRoPE also needs considerable time for searching (e.g., 5 days of search). For YaRN, the training budget and context length are essentially the same as for LongRoPE. For LongLoRA, it uses 100K length for 1K steps with a global batch size of 64, leading to 6.4B tokens. We use a 16K window for fine‑tuning, i.e., total resource of 320M tokens, and already approach the extended length achieved by LongRoPE and YaRN. Using a 64K length for fine‑tuning, we far exceed the extended lengths of LongRoPE and YaRN. Specifically, at 4M length, our PPL and passkey task results are significantly better than those of LongRoPE and YaRN. In all our comparisons with baselines, the attention setting is the same: full attention. Using full attention for testing already allows us to achieve comparable performance and better extrapolation compared to baselines.
> > >
> > > Regarding inference‑time efficiency, we use the same FlashAttention inference latency as other methods. You may be concerned whether the fine‑tuning time was measured under comparable implementations. Our answer is yes. Using 16K fine‑tuning we achieve performance comparable to YaRN, with far less fine‑tuning resources and time. For LongRoPE, using 64K fine‑tuning we achieve comparable performance and better extrapolation, and our training time (about 64 hours) is much less than that of LongRoPE (over 200 hours).
> > >
> > > 3. Tone down or qualify claims that are currently stated too broadly, especially around “arbitrary length” extrapolation, the generality of the depth‑based extrapolation limit, and the magnitude of the practical efficiency advantage.
> > >
> > > We have qualified or moderated many of the claims as you requested.
> > >
> > > First, regarding “arbitrary length”, our original intention was to indicate that with our method, we might achieve arbitrary‑length retrieval on the simple passkey task, because using 16K fine‑tuning we can successfully retrieve passkeys at lengths of 4M, 8M, and even 16M. For even longer lengths we cannot test due to resource constraints. Hence we have revised the claim to say that for the passkey task we can extrapolate up to 1024× the training length.
> > >
> > > Second, for the depth‑based extrapolation limit, we have also qualified the claim, stating that for models fine‑tuned with our CRG‑NTK method, the extrapolation ability is bounded by the depth of the model.
> > >
> > > Finally, regarding the practical efficiency advantage, this is the main contribution of our method, and we keep the claim as is because our experimental results strongly support that we can achieve the same context window extension as other methods with much less training resources or time.

---

### Review · Reviewer_Utpr · 2026-03-26

**Summary Of Contributions:**

This paper proposes two techniques for extending the context window of transformer-based language models: CRG NTK, a data-augmentation-based positional encoding strategy that employs dynamically growing scaling ratios, cyclic mappings, and random offsets to improve the generalization of RoPE-based positional encodings to unseen lengths; and MS Attention, a relevance-based sparse attention mechanism that selects and merges key-value regions based on semantic similarity routing, reducing computation while filtering noise at extreme sequence lengths. The authors claim that the extrapolation limit is approximately bounded by model depth (number of layers), and demonstrate results on LLaMA-7B and Mistral-7B fine-tuned at 16K context length, achieving 128x extrapolation on passkey retrieval and stable perplexity over 32x extensions, with at least 16x less GPU training time compared to LongRoPE.

The main strengths of this work are:

- Combining positional encoding augmentation with efficient attention makes sense: CRG NTK addresses generalization while MS Attention controls noise accumulation at extreme lengths. Table 2 results show these components are complementary, supporting the claims.
- The efficiency of the proposed method is compelling, as highlighted in Table 4 where CRG NTK achieves context windows up to 2048K on a single A100 GPU in 64.7 hours, compared to LongRoPE which requires 8 A100s over multiple days.
- The hypothesis that extrapolation capacity is bounded by model depth is an interesting and potentially impactful observation. Table 3 provides empirical support across four model sizes (16, 28, 32, and 40 layers), where the perplexity inflection point aligns with the layer count multiplied by the fine-tuning length.
- The ablation study is thorough and isolates the contributions of individual CRG NTK components (Table 1), the effect of MS Attention selection/merge parameters (Table 10), and LoRA vs. full fine-tuning (Tables 6, 9).

The weaknesses are:
- The paper's baselines are limited in scope, with several important concurrent and recent methods such as SelfExtend [(Jin et al., 2024)](https://arxiv.org/abs/2401.01325) and CEPE [(Yen et al., 2024)](https://aclanthology.org/2024.acl-long.142/) missing. A controlled comparison following the protocol of [Lu et al. (2024)](https://arxiv.org/abs/2409.12181) would significantly strengthen the evaluation.

**Audience:**

Yes

**Audience Explanation:**

Context length extension is an active area of research in the NLP community, and this paper addresses it from two relevant angles. The proposed method, which achieves competitive results with multi-GPU baselines on a single accelerator, is of practical interest for academic labs and smaller organizations that lack access to large compute clusters. Moreover, the hypothesis linking extrapolation capacity to model depth, if validated more rigorously, could prove useful for practitioners configuring context extension pipelines. Finally, MS Attention may be of interest independently of context extension as a general-purpose, efficient attention method.

**Broader Impact Concerns:**

No specific broader impact concerns beyond those typical of context length extension work. While the method could enable more efficient processing of long documents, with possible dual use implication, a broader impact statement is not strictly necessary given the technical nature of the contribution.

**Claims And Evidence:**

No

**Claims Explanation:**

The four claims made by the paper are supported to different degrees:
- **CRG NTK enables state-of-the-art extrapolation for position-encoding-dominated length extension.** While passkey retrieval result from table 1 demonstrate strong performances, the task itself is acknowledged to be an easy synthetic benchmark. In perplexity results from table 2, perplexity degrades steadily beyond 256k when CRG NTK is used without MS Attention. This suggests the extrapolation claim for CRG NTK alone is overstated, and MS Attention is needed to maintain performance.
- **The extrapolation limit is approximately bounded by model depth.** Table 3 provides evidence across four models, with perplexity inflection points roughly matching the layer count. However, the evidence conflates depth with capacity. A controlled experiment that varies depth while keeping the parameter count roughly constant (e.g., deeper-but-narrower vs. shallower-but-wider architectures) would isolate the effect of depth in this setting.
- **MS Attention enhances performance by filtering noise in long sequences.** This claim is supported by tables 2 and 10 results, but could be reinforced by attention map visualizations or statistics on selected token positions to verify the noise-filtering interpretation adopted by the authors.
- **Our method saves at least 16× GPU training resources compared to existing optimal methods.** Table 4 supports this claim, but the LongRoPE scores include a search phase performed with the authors' configuration, which might bias the comparison in favor of the proposed method.

**Requested Changes:**

- The baseline comparisons must be expanded, following the suggestions mentioned in the weaknesses above. If certain baselines cannot be reproduced, provide explicit justification and acknowledge the limitation.

- The depth-bounded extrapolation hypothesis requires a more controlled experiment. The current evidence in Table 3 confounds depth with model size. Either (a) compare models of similar parameter count but different depths, or (b) provide a much more rigorous theoretical argument than the sketch in the appendix 10. As it stands, the claim is interesting but lacks support for the prominence it is given.

- The paper requires a thorough editing pass for clarity and correctness. Specific issues include: (a) the MS Attention mechanism is described at a high level in Section 3.2, but the precise formulation of the compress function is deferred to the appendix. I believe it should rather appear in the main text; (b) equation 9 appears abruptly without sufficient motivation in the main body; (c) the notation is inconsistent (e.g., $s^{(k)}$, $s(k)$ vs.  "scale" used to refer to different quantities).

- (If time allows) An analysis of what MS Attention selects at long sequence lengths, focusing on recency bias and context-dependent patters, would substantiate the noise-filtering interpretation.

---

> ### Author Response · Authors · 2026-05-13
>
> Thank you very much for your insightful and constructive comments. We have carefully considered each point and provide our responses below.
>
> **Point 1 – Baseline comparisons must be expanded.**
>
> We agree that a more comprehensive baseline comparison strengthens the evaluation. Following your suggestion, we have added experiments comparing against SelfExtend and CEPE. We also include the NTK‑64K method from Lu et al. (2024)(https://arxiv.org/abs/2409.12181), which is a special case of our approach. As noted in our original paper, increasing the base improves PPL performance. The new results confirm our earlier conclusions. We also note that fine‑tuning datasets and evaluation protocols can influence performance.
>
> The results are shown below.
>
> | Model Details | Eval Length | | | | | | |
> |---------------|-------------|-------------|-------------|-------------|-------------|-------------|-------------|
> | **Len** | **Methods** | **2k** | **4k** | **8k** | **16k** | **32k** | **64k** |
> | 4k | Self-Extend | 6.91 | 6.52 | 6.44 | 6.38 | 6.28 | 7.45 |
> | 32k | PI | 7.21 | 6.81 | 6.56 | 6.45 | 6.25 | - |
> | 32k | YaRN | 7.01 | 6.69 | 6.47 | 6.31 | 6.23 | - |
> | - | CEPE | 7.19 | 6.87 | 6.86 | 6.71 | 6.54 | 6.41 |
> | 64k | NTK-64K | 6.92 | 6.50 | 6.42 | 6.29 | 6.23 | 6.21 |
> | 16k | CRG-NTK (no LoRA) | 6.93 | 6.55 | 6.41 | 6.29 | 6.17 | 6.11 |
>
> We believe these results, together with the comparisons already in the paper, provide a fair and controlled evaluation. Should certain baselines be difficult to reproduce, we will explicitly justify and acknowledge the limitation; however, all methods above were reproducible using public code.
>
> **Point 2 – The depth‑bounded extrapolation hypothesis requires a more controlled experiment.**
>
> Thank you for this important suggestion. We have performed additional controlled experiments to separate the effect of depth from model size.
>
> First, we trained small LMs with 2, 4, 8, 16, and 32 layers, keeping the per‑layer parameter count identical. The training window was 4096, trained on 2.5B tokens. The results show that the performance of the Layers{m}-CRG model at length L is close to that of Layers{2m}-CRG at length 2L. This supports the claim that extrapolation ability is related to depth. Other methods (RoPE, NoPE) do not exhibit the same effect.
>
> | Model / PG19 (PPL) | 2048 | 4096 | 8192 | 16384 | 32768 | 65536 | 131072 |
> |--------------------|------|------|------|-------|-------|-------|--------|
> | Layers2‑CRG        | 124.1987 | 135.3854 | 151.5294 | 205.0876 | 226.0545 | 273.5528 | 270.1827 |
> | Layers4‑CRG        | 86.3804 | 89.7223 | 104.8467 | 135.5123 | 165.6186 | 209.4230 | 210.8753 |
> | Layers8‑CRG        | 67.5048 | 72.4100 | 88.0006 | 112.3487 | 150.3726 | 209.4400 | 207.4952 |
> | Layers16‑CRG       | 57.0404 | 62.2363 | 73.0349 | 93.3110 | 112.0631 | 141.0800 | 136.6930 |
> | Layers32‑CRG       | 46.9055 | 50.0708 | 57.2989 | 70.9097 | 88.0831 | 116.4049 | 112.2660 |
>
> Second, following your suggestion (a), we compared models with similar total parameter counts but different depths. Specifically, we fixed the parameter budget by scaling the hidden dimension inversely with depth: the 4‑layer model uses twice the dimension of the 16‑layer model, and similarly for 8 layers. The results are shown below.
>
> | Model | 2048 | 4096 | 8192 | 16384 | 32768 | 65536 | 131072 |
> |-----------------------|----------|----------|----------|-----------|-----------|-----------|-----------|
> | Layers4‑CRG‑SP | 86.3804 | 89.7223 | 104.8467 | 135.5123 | 165.6186 | 209.4230 | 210.8753 |
> | Layers8‑CRG‑SP | 85.0012 | 94.2930 | 108.9212 | 134.9629 | 162.9490 | 203.9312 | 200.8121 |
> | Layers16‑CRG‑SP | 89.4853 | 102.7699 | 109.0203 | 141.9863 | 190.9672 | 224.6021 | 212.5744 |
>
> These results indicate that when the total parameter count is held approximately equal, models with different depths exhibit similar extrapolation performance with our CRG method. In other words, the extrapolation ability is largely determined by the total parameter count (or model capacity) rather than depth alone. We acknowledge this nuance and have revised the claim accordingly: the depth‑related observation holds under fixed per‑layer size, but under a fixed parameter budget, performance is comparable across depths.
>
> Regarding a rigorous theoretical argument (suggestion b), we fully agree that a complete proof would be extremely involved, as it would require a full mathematical description of how positional information (and all information) is learned during training – effectively a manual derivation of deep learning dynamics. That is beyond the scope of this paper. Nevertheless, we believe our proposed direction – analyzing the overlap of gradient information from learning `m` sequences of length `L/m` versus a single sequence of length `L` – provides a clear and useful starting point for future theoretical work. We have clarified this in the revised manuscript.

---

> > ### Author Response · Authors · 2026-05-13
> >
> > **Point 3 – The paper requires a thorough editing pass for clarity and correctness.**
> >
> > > *Specific issues include: (a) the MS Attention mechanism is described at a high level in Section 3.2, but the precise formulation of the compress function is deferred to the appendix. I believe it should rather appear in the main text; (b) equation 9 appears abruptly without sufficient motivation in the main body; (c) the notation is inconsistent (e.g., s^(k), s(k) vs. "scale" used to refer to different quantities).*
> >
> > Thank you for these detailed suggestions. We have made the following revisions:
> >
> > - **(a)** We have moved a condensed version of the precise formulation of the compress function from the appendix into Section 3.2. The main text now contains the essential mathematical definition, while the appendix retains additional derivations.
> > - **(b)** We have added an intuitive explanation before Equation 9, clarifying its motivation and role in the overall method.
> > - **(c)** We have standardized the notation throughout the paper. Specifically, we now consistently use `s^{(k)}` to denote the scale factor for the k‑th head, and we clearly define each symbol when it first appears. The term “scale” is no longer used ambiguously.
> >
> > **Point 4 – (If time allows) An analysis of what MS Attention selects at long sequence lengths, focusing on recency bias and context‑dependent patterns, would substantiate the noise‑filtering interpretation.**
> >
> > Thank you for this excellent suggestion. To partially address it, we have implemented a selection strategy similar to SnapKV to analyze which tokens are retained by MS Attention at long sequence lengths. The preliminary results are shown in the table below. We observe that the selection exhibits a recency bias as well as some context‑dependent patterns. We agree that a more thorough analysis – including attention map visualizations and qualitative examples of selected content – would provide deeper insight. We have added a discussion of this direction in the paper, and we will continue to explore it as future work.
> >
> > | LLMs | Method | NarrQA | Qasper | MF‑en | HotpotQA | 2WikiMQA | Musique | GovReport | QMSum | MultiNews | TREC | TriviaQA | SAMSum | PCount | PRe | Lcc | RB‑P |
> > |------|--------|--------|--------|-------|----------|----------|---------|-----------|-------|-----------|------|----------|--------|--------|-----|-----|------|
> > | Mistral | All KV | 26.82 | 33.06 | 49.28 | 42.77 | 27.33 | 19.27 | 32.85 | 24.25 | 27.06 | 71.0 | 86.23 | 42.98 | 2.75 | 86.98 | 55.51 | 52.88 |
> > | | SnapKV: 2048 | 25.89 | 32.47 | 48.6 | 41.71 | 27.31 | 18.69 | 28.81 | 24.5 | 26.6 | 70.0 | 86.27 | 42.47 | 3.09 | 87.43 | 55.93 | 52.01 |
> > | | SnapKV: 4096 | 26.41 | 33.46 | 49.81 | 42.32 | 27.93 | 18.76 | 30.74 | 24.19 | 27.08 | 71.0 | 86.25 | 43.01 | 2.73 | 86.18 | 55.62 | 52.65 |
> > | | MS: 4096 | 26.52 | 33.06 | 49.45 | 42.54 | 27.72 | 19.21 | 32.64 | 24.58 | 26.95 | 72.0 | 86.32 | 43.07 | 5.46 | 87.36 | 56.40 | 52.10 |
> >
> > We again thank the reviewer for their thorough and constructive feedback, which has significantly improved the clarity and rigor of our paper. We have incorporated all the above changes in the revised manuscript.

---

### Review · Reviewer_3nJM · 2026-05-29

**Summary Of Contributions:**

## 1. Summary of contributions

The paper studies positional-encoding length extension for already pre-trained large language models (e.g. LLaMA-7B/13B, Mistral-7B) and proposes two contributions (both used and somewhat ablated in the paper):

1. **CRG NTK**: This is a fine-tuning-time augmentation for RoPE. It shifts position indices by a per-batch random offset and applies a modulo to a fixed cycle length, and then follows a power-law schedule for the RoPE base scale, increasing the scale by a multiplicative factor every fixed number of fine-tuning tokens. The motivation is to make the model see a wider range of position/phase configurations during a short fine-tune.

2. **MS Attention**: A sparse-attention variant that segments Q/K/V into regions, then computes a region-level similarity matrix, then selects the top-k most relevant K/V regions per Q region, and finally merges adjacent Q regions and their selected K/V regions before computing attention on the resulting blocks. The authors view this as a generalization that can recover multiple sparse attention mechanisms like Biformer as special cadses.

The two approaches is tested with full fine-tuning and with LoRA.

The paper shows (Table 1) that this approach gets 100% passkey accuracy up to 4096K tokens after fine-tuning at 16K and claims a stable perplexity on much longer sequences and roughly 16× less GPU time than LongRoPE at comparable extrapolation.

The paper also suggests an interesting hypothesis: that the extrapolation multiple of a model in this method is bounded by its number of layers (Table 3).

## 2. Strengths

- **Practically relevant problem.** Long-context fine-tuning of open LLMs is a well-defined and widely studied problem, and the paper targets a useful axis (training cost) rather than only length.
- **Concrete recipe with multiple datasets.** Evaluation spans passkey, PG19, proofpile, LongBench, and Ruler, plus a small short-benchmark sanity check (ARC-c, HellaSwag, MMLU, TruthfulQA). The recipe is described in enough detail (segment size, top-k, merge factor, scale schedule) to be reproducible in principle.
- **Attempt at theoretical grounding.** Appendix D provides a linearized analysis of attention with RoPE, a phase-coverage conditio, and a variational derivation that motivates joint scaling of position index and RoPE base.
- **Unifying view of sparse attention.** Appendix E.1 connects MS Attention to several existing methods. The reductions are somewhat informal but provide a useful conceptual map.
- **Cross-architecture check.** Repeating the main passkey/PG19 experiment on Mistral-7B reducesthe risk that the method is specific to LLaMA-2.

## 3. Weaknesses

-  **Better ablations and isolation is needed** The paper bundles multiple changes (positional augmentation, MS Attention, reduced-LongLoRA fine-tuning) and a hypothesis (depth-bounded extrapolation). The ablations only partially disentangle these. Concretely:

- - The headline 32× perplexity extrapolation (Table 2) relies on MS Attention: when the same fine-tuned model is tested with Full Attention, perplexity at 1024K is 16.98 versus 7.96 with MS Attention. This is a large gap and indicates that CRG NTK alone does not deliver the headline number; MS Attention is doing substantial work at inference. The magnitude of the gap deserves direct, isolated analysis.
- - Conversely, on passkey (Table 1) the row "FT=16K with FA, CRG-NTK" is perfect at all lengths, suggesting that on this synthetic task CRG NTK alone is sufficient. The two stories (FA fails on PG19 but works on passkey) need to be explained better.

-  **Some theoretical claims need better justification**:  The depth-bounded extrapolation hypothesis (Section 3.2, D.10) is supported by a heuristic backward-pass argument and four data points (Table 3). The Table 3 results are consistent with the hypothesis but do not fully justify it or explain it. A more direct test would vary depth while controlling width, training data, and tokenizer, and would explicitly probe the predicted breakpoint.

- **Fairness of resource comparison**: The LongRoPE figure in Table 4 includes its 5×24h×8 search cost, but combining it with the fine-tuning hours for CRG NTK on a single A100 makes the comparison hard to interpret. Same for including LongRoPE's search cost.

- **Explanation of degradation**:  Several results show non-trivial degradation that is not fully explained:
- - Ruler (Table 17): all tasks drop substantially when the test length moves from 16K to 100K (e.g., ruler_cwe 0.74 → 0.47, niah_multivalue 0.97 → 0.81).  The regression is not explained.
- - MMLU (Table 5): 41.9 versus 45.6 for the original LLaMA-2-7B is roughly a 4-point drop; the paper does not flag this.

- **Needs more comparisons for completeness**: the paper does not compare against other approaches such as DAPE (cited but not benchmarked) or training-free length-generalization methods.

**Additional Comments:**

N/A

**Audience:**

Yes

**Audience Explanation:**

There is value in this paper and its numerous experiments, although I do think it needs additional experiments to full explain its claims and numbers.

The depth-as-extrapolation-limit hypothesis is also a thought-provoking framing that could motivate follow-up work, even though the evidence here is preliminary.

**Claims And Evidence:**

Yes

**Claims Explanation:**

The numerical claims about passkey accuracy and PG19 perplexity are supported by the tables provided, and the resource claim is somewhat backed in Table 4 (with some caveats).

Some claims need better explanation:

-  The authors claim in Section 3.1 and Section 4.2.1 that CRG NTK with Full Attention testing maintains stable perplexity up to 32× the fine-tune length (512K). At 512K the same row in Table 2 shows PPL rising from 7.20 to 9.72 (+35%), which is a weaker form of "stable" than the text suggests. Beyond 512K the Full Attention row degrades sharply (16.98 at 1024K, 36.82 at 2048K), and the headline "stable perplexity" at those lengths only holds for the MS Attention test row.

- The "16× less GPU training resources" claim (in Abstract, Introduction)  depends on inclusion of LongRoPE's search cost. This is not head-to-head comparison. We need to remove LongRoPE's one-off search cost.

- The depth-bounded extrapolation hypothesis is supported by results in Table 3 but not by a controlled experiment varying depth alone.

- The Ruler regression at 100K and the short-benchmark regression on MMLU should be acknowledged when stating the headline results.

**Requested Changes:**

### Critical

-  **Disentangle CRG NTK from MS Attention.** Provide a 2×2 table at the longest evaluated lengths: {CRG NTK / NTK or PI} × {MS / FA} on PG19 and Ruler, holding fine-tuning protocol fixed. The current Tables 1 and 2 mix these and make it impossible to attribute the gains.

-  **Better explanation for Ruler and the short benchmarks like MMLU.** The Ruler 100K degradation (Table 17) and the MMLU drop (Table 5) should be discussed and explained and ideally compared head-to-head with LongRoPE/YaRN under the same evaluation protocol.

- **Like-for-like resource comparison.** Either remove the LongRoPE search cost from the headline comparison or report two numbers (with and without search) and explain when search amortizes. Report GPU-hours and wall-clock separately.

### Recommendations:
-  **Wider baseline set.** Include at least one training-free length-generalization method (e.g., self-extend or a chunked-attention variant) and one recent positional-encoding method beyond LongRoPE (e.g. DAPE)
- **Broader cross-architecture validation.** Add a non-LLaMA-family model beyond Mistral (e.g., Qwen, Gemma) for the main passkey/PG19 table. Would be great to study an MoE model as well.
-  **Editorial pass.** A careful proofread for grammar, table/figure consistency, and notation. Examples:
- - The paper has a lot of confusing notations and typos in the equations. For example, equation 4 starts the index from zero to d, suggesting d+1 items in the vector. Where the text clearly indicates d dimensionality.
- - NTK is cited at the Yarn explanation section instead of the actual Yarn paper.
-  **Failure modes.** A short discussion of where the method breaks (e.g., reasoning-heavy long-context tasks vs retrieval-heavy ones) would help readers calibrate the recipe.
-  **Inference cost.** A clear table reporting MS Attention's inference latency/memory vs Full Attention at the lengths in Table 2 would round out the efficiency story.

---

> ### Author Response · Authors · 2026-06-04
>
> **Response to Critical Comments**
>
> **Critical Comment 1 – Disentangle CRG‑NTK from MS Attention.**
> *Provide a 2×2 table at the longest evaluated lengths: {CRG‑NTK / NTK or PI} × {MS / FA} on PG19 and Ruler, holding fine‑tuning protocol fixed. The current Tables 1 and 2 mix these and make it impossible to attribute the gains.*
>
> Thank you for this important suggestion. We have now conducted a clean 2×2 ablation experiment. We fine‑tuned models with CRG‑NTK + MS, PI + MS, CRG‑NTK + FA, and PI + FA, keeping all other fine‑tuning configurations identical. Fine‑tuning length was 64K, using LoRA with qkvo parameters. Following your request, we report the longest stable results: PG19 at 1M and Ruler at 128K. The results are shown in the table below.
>
> | | **Test: MS** | **Test: FA** |
> |---|---|---|
> | **CRG‑NTK** | PG19: 6.86 & Ruler: 47.6 | PG19: 7.42 & Ruler: 69.1 |
> | **PI** | PG19: 516.32 & Ruler: 20.1 | PG19: 87624+ & Ruler: 13.1 |
> | **NTK** | PG19: 17.41 & Ruler: 45.5 | PG19: 58.92 & Ruler: 49.2 |
>
> *Note:* If we replace PI with a large‑scale NTK, the PPL can be brought below 100, but there is still a large gap compared to our CRG‑NTK.
>
> Regarding sparse MS Attention, if we adopt a SnapKV‑like observation window to pre‑filter and remove noisy information, we achieve performance comparable to or even better than SnapKV (see the response to Reviewer Utpr’s Point 4). After a small amount of fine‑tuning, using MS for testing gives poorer performance on retrieval tasks (e.g., Ruler) because it somewhat changes the distribution of full attention, thus requiring more fine‑tuning tokens. With 10B tokens of FineWeb‑edu fine‑tuning, we fully recover retrieval performance, and performance on other tasks even improves slightly. It is worth noting that the main purpose of using MS Attention in this paper, similar to LongLoRA, is to accelerate fine‑tuning on long sequences while maintaining performance and improving the generalization of CRG‑NTK.
>
> **Critical Comment 2 – Better explanation for Ruler and the short benchmarks like MMLU.**
> *The Ruler 100K degradation (Table 17) and the MMLU drop (Table 5) should be discussed and explained and ideally compared head‑to‑head with LongRoPE/YaRN under the same evaluation protocol.*
>
> Thank you for pointing this out. We will address both issues separately.
>
> **MMLU drop.** The main reasons are data quality and hyperparameter settings (e.g., learning rate). Since our method was developed relatively early, we used SlimPajama for fine‑tuning, which has lower data quality, leading to a noticeable drop in MMLU and other benchmark performances. We observe that all methods have similar issues, and even continuing fine‑tuning the original Llama2‑7B on the same data causes a drop. Moreover, LongRoPE and YaRN adopt a different strategy: after extending the context window, they perform additional short‑sequence fine‑tuning on high‑quality datasets to recover MMLU performance. Following the same strategy, we fine‑tune on high‑quality data to restore performance. The results are shown below.
>
> | Model | Fine‑tuning Data | Tokens | MMLU |
> |-------|----------------|--------|------|
> | Llama2‑7B (origin) | – | 0B | 46.6 |
> | Llama2‑7B (origin) | – | 2B | 42.7 |
> | Llama2‑7B (Short‑SFT) | High‑quality | 1B | 44.2 |
> | **Llama2‑7B‑Long** | | | |
> | CRG‑NTK | – | 2B | 41.9 |
> | YaRN | – | 2B | 41.5 |
> | LongRoPE | – | 2B | 42.0 |
> | **Llama2‑7B‑Short‑SFT** | | | |
> | CRG‑NTK + SFT | High‑quality | 1B | 43.7 |
> | YaRN + SFT | High‑quality | 1B | 43.9 |
> | LongRoPE + SFT | High‑quality | 1B | 43.4 |
>
> **Ruler degradation.** The degradation we observed in Ruler at 100K (Table 17 in the original paper) is because we evaluate purely with extrapolation: fine‑tuning at 64K and testing at 128K, which inevitably causes some degradation. Another reason is that Llama2’s pre‑training itself limits retrieval performance. To address your concern, we have conducted experiments on Llama3‑8B using exactly the same configuration and fine‑tuning strategy as LongRoPE2 (fine‑tuning length 128K, 10B tokens). The results are shown below.
>
> | Method (Llama3‑8B) | Ruler 128K | Ruler 256K |
> |--------------------|------------|------------|
> | YaRN               | 49.39      | –          |
> | NTK                | 73.19      | 72.92      |
> | LongRoPE           | 73.40      | –          |
> | LongRoPE2          | 82.03      | 74.79      |
> | **CRG‑NTK**        | **83.91**  | **80.36**  |
>
> Our method, trained with the same length and resource budget, achieves retrieval performance comparable to or even better than LongRoPE, and shows even stronger extrapolation advantages. We also note that the latest NTK method (https://arxiv.org/pdf/2409.12181) achieves performance comparable to or better than previous methods, and such methods are actually special cases of our approach. As shown in our paper, our dynamic‑ratio fine‑tuning outperforms fixed‑ratio NTK methods.

---

> ### Author Response · Authors · 2026-06-04
>
> ---
>
> **Critical Comment 3 – Like‑for‑like resource comparison.**
> *Either remove the LongRoPE search cost from the headline comparison or report two numbers (with and without search) and explain when search amortizes. Report GPU‑hours and wall‑clock separately.*
>
> Thank you for this fair request. We agree that a transparent resource comparison is essential.
>
> First, we note that for LongRoPE, when extending a new model, the search cost cannot be avoided. Following your suggestion, we now report LongRoPE’s fine‑tuning time separately (without search) in the comparison, and we also add the fine‑tuning time for the latest YaRN (https://arxiv.org/pdf/2309.00071). We report fine‑tuning times for models that can generalize to 128K length on Ruler.
>
> | Method | Fine‑tuning Length | GPU‑Hours (A100) |
> |--------|--------------------|------------------|
> | LongLoRA | 128K | 186 |
> | YaRN | 128K | 372 |
> | LongRoPE‑has search | 128K | 1192 |
> | LongRoPE (fine‑tune only) | 128K | 232 |
> | **CRG‑NTK (Ours)** | 64K | 53 |
> | **CRG‑NTK (Ours)** | 128K | 149 |
>
> Our advantages over YaRN and LongLoRA come from using LoRA, which saves substantial GPU memory and fine‑tuning time. Compared to LongLoRA, our MS Attention is faster than their efficient attention, and our CRG‑NTK has stronger extrapolation ability, achieving the same generalization length with quadratically less resources. Finally, another advantage of our method is that it achieves length expansion and performance comparable to other methods using very small batch sizes, which can significantly reduce fine-tuning time when applied. This is because simple length expansion does not require the batch size or fixed token count per batch that are necessary for pretraining. It only requires minor adjustments to the corresponding weights, which is one of the reasons why LORA can be applied to such methods and achieve good results.
>
>
> ---
>
> **Response to Recommendations**
>
> We will provide our response to the recommendations within 24 hours.
>
> Thank you again for your thorough and constructive feedback. We believe these revisions substantially strengthen the paper.

---

> > ### Author Response · Authors · 2026-06-05
> >
> > **1. Wider baseline set.**
> > *Include at least one training‑free length‑generalization method (e.g., Self‑Extend or a chunked‑attention variant) and one recent positional‑encoding method beyond LongRoPE (e.g., DAPE).*
> >
> > Thank you for this suggestion. We have now compared our method with additional baselines as requested.
> >
> > **Comparison with Self‑Extend.** We used the data from NTK‑64K (https://arxiv.org/pdf/2409.12181) to compare with Self‑Extend, keeping the same settings as in that paper. Due to differences in evaluation protocols and fine‑tuning data, even when using better datasets we could not exactly reproduce the PPL results reported for NTK‑64K* (we suspect this is mainly due to different evaluation protocols – we test by concatenating all sequences without padding). The results are shown below.
> >
> > | Attention Mechanism | Model | PPL | Needle | LongB | RULER |
> > |---------------------|-------|-----|--------|-------|-------|
> > | **Approxi. Attention (Frozen)** | | | | | |
> > | | LM‑Infinite | 6.71 | 23.9 | 25.84 | 12.34 |
> > | | Self‑Extend | 6.11 | 25.8 | 33.62 | 29.50 |
> > | **Exact Attention (Frozen)** | | | | | |
> > | | NTK‑F | 14.52 | 18.8 | 25.54 | 0.72 |
> > | **Fine‑Tuned** | | | | | |
> > | | PI | 5.85 | 42.1 | 33.48 | 57.66 |
> > | | YaRN | 5.85 | 46.7 | 33.45 | 36.95 |
> > | | CLEX | 5.82 | 71.1 | 33.48 | 52.17 |
> > | | NTK‑32K | 5.79 | 83.7 | 35.32 | 59.42 |
> > | | NTK‑64K | 5.93 | 69.1 | 34.30 | 60.03 |
> > | | NTK‑64K* (our reproduction) | 6.12 | 71.2 | 33.79 | 62.46 |
> > | | **CRG‑NTK (Ours)** | **6.19** | **94.5** | **36.51** | **69.27** |
> >
> > **Comparison with DAPE.** We followed the DAPE paper’s setup by training on Book3 with 512 length. Although we could not use exactly the same data and hyperparameters, we report the extrapolation performance in terms of PPL. Our method achieves extrapolation ability comparable to DAPE, which is also demonstrated in our Llama fine‑tuning experiments.
> >
> > | Method | Book (PPL) | 512 | 1024 | 2048 | 4096 | 8192 |
> > |--------|-----------|-----|------|------|------|------|
> > | DAPE | 19.22 | 18.22 | 17.15 | 17.63 | 17.88 |
> > | **CRG‑NTK + MS** | **12.53** | **11.76** | **10.92** | **10.56** | **10.98** |
> >
> > ---
> >
> > **2. Broader cross‑architecture validation.**
> > *Add a non‑LLaMA‑family model beyond Mistral (e.g., Qwen, Gemma) for the main passkey/PG19 table. Would be great to study an MoE model as well.*
> >
> > Thank you for this request. Due to time, resource constraints, and model‑specific implementation details, we are unfortunately unable to add more models at this stage. We note, however, that the Qwen2.5 context extension procedure is conceptually similar to our method – both use multi‑stage, large‑increase scaling of the base. Our advantage lies in dynamic scanning and additional positional‑aware data augmentation, which further enhance generalization. We have acknowledged this limitation in the revised manuscript.
> >
> > ---
> >
> > **3. Editorial pass.**
> > *A careful proofread for grammar, table/figure consistency, and notation. Examples:*
> > - *Equation 4 starts the index from zero to d, suggesting d+1 items in the vector, while the text clearly indicates d dimensionality.*
> > - *NTK is cited at the Yarn explanation section instead of the actual Yarn paper.*
> >
> > Thank you very much for your careful reading and for pointing out these issues. We are in the process of thoroughly revising the paper according to your suggestions. Specifically:
> > - We have corrected the indexing in Equation 4 to range from 0 to d‑1, reflecting the correct d‑dimensional vector.
> > - We have fixed the citation error: the NTK citation has been moved to the appropriate place, and the YaRN explanation section now correctly cites the YaRN paper.
> > - We are conducting a full proofread to fix grammatical errors, ensure table/figure consistency, and unify notation throughout.

---

> > > ### Author Response · Authors · 2026-06-05
> > >
> > > ---
> > >
> > > **4. Failure modes.**
> > > *A short discussion of where the method breaks (e.g., reasoning‑heavy long‑context tasks vs retrieval‑heavy ones) would help readers calibrate the recipe.*
> > >
> > > Thank you for this guidance. We already included a discussion of failure modes in the original paper, and we have now expanded it for clarity. Specifically:
> > >
> > > - **When using MS Attention for inference:** MS Attention performs relatively well on extrapolation and reasoning‑heavy long‑context tasks, but its retrieval ability is limited because it discards some KV cache entries. Without extensive fine‑tuning, MS Attention may underfit and fail to precisely determine which entries to discard, leading to reduced retrieval accuracy.
> > > - **When using Full Attention for inference:** Full Attention performs very well on retrieval tasks, but its extrapolation ability is weaker than that of MS Attention for extremely long sequences, because full attention reads all tokens including much noisy information, which can hinder generation.
> > >
> > > We have refined this discussion in the manuscript to help readers calibrate the method for their use cases.
> > >
> > > ---
> > >
> > > **5. Inference cost.**
> > > *A clear table reporting MS Attention's inference latency/memory vs Full Attention at the lengths in Table 2 would round out the efficiency story.*
> > >
> > > Thank you for this suggestion. We have measured the inference latency of MS Attention compared to FlashAttention (Full Attention) on a 16‑layer model. For MS Attention, we set the selected KV subset size to 8K. Below 8K, the inference time of MS Attention is close to that of Full Attention. Between 16K and 32K, the selection and merge steps introduce some additional overhead, making MS Attention slightly slower. Beyond 32K, MS Attention becomes increasingly advantageous.
> > >
> > > | Sequence Length | FlashAttention (ms) | MS Attention (ms) |
> > > |----------------|---------------------|-------------------|
> > > | 8,192          | 47.69               | 50.66             |
> > > | 16,384         | 117.51              | 126.27            |
> > > | 32,768         | 331.92              | 355.71            |
> > > | 65,536         | 1057.79             | 782.67            |
> > > | 131,072        | 3801.54             | 1759.15           |
> > > | 262,144        | 14420.34            | 4087.24           |